# Lysine demethylase 7a regulates murine anterior-posterior development by modulating the transcription of *Hox* gene cluster

Yoshiki Higashijima [1,2,9], Nao Nagai[3], Masamichi Yamamoto[4,10], Taro Kitazawa[5,6], Yumiko K. Kawamura[5,6], Akashi Taguchi[2], Natsuko Nakada[2], Masaomi Nangaku[7], Tetsushi Furukawa[1], Hiroyuki Aburatani[8], Hiroki Kurihara[6], Youichiro Wada[2] & Yasuharu Kanki [2,11✉]

Temporal and spatial colinear expression of the *Hox* genes determines the specification of positional identities during vertebrate development. Post-translational modifications of histones contribute to transcriptional regulation. Lysine demethylase 7A (Kdm7a) demethylates lysine 9 or 27 di-methylation of histone H3 (H3K9me2, H3K27me2) and participates in the transcriptional activation of developmental genes. However, the role of Kdm7a during mouse embryonic development remains to be elucidated. Herein, we show that *Kdm7a*$^{-/-}$ mouse exhibits an anterior homeotic transformation of the axial skeleton, including an increased number of presacral elements. Importantly, posterior *Hox* genes (caudally from *Hox9*) are specifically downregulated in the *Kdm7a*$^{-/-}$ embryo, which correlates with increased levels of H3K9me2, not H3K27me2. These observations suggest that Kdm7a controls the transcription of posterior *Hox* genes, likely via its demethylating activity, and thereby regulating the murine anterior-posterior development. Such epigenetic regulatory mechanisms may be harnessed for proper control of coordinate body patterning in vertebrates.

[1] Department of Bioinformational Pharmacology, Tokyo Medical and Dental University, Tokyo 113-8510, Japan. [2] Isotope Science Center, The University of Tokyo, Tokyo 113-0032, Japan. [3] Department of Microbiology and Immunology, Keio University School of Medicine, Tokyo 160-8582, Japan. [4] Department of Nephrology, Kyoto University Graduate School of Medicine, Kyoto University Hospital, Shogoin-Kawaramachi-cho, Sakyo-ward, Kyoto 606-8507, Japan. [5] Friedrich Miescher Institute for Biomedical Research, Basel 4051, Switzerland. [6] Department of Physiological Chemistry and Metabolism, The University of Tokyo, Tokyo 113-0033, Japan. [7] Division of Nephrology and Endocrinology, The University of Tokyo, 113-0033 Tokyo, Japan. [8] Division of Genome Science, RCAST, The University of Tokyo, Tokyo 153-8904, Japan. [9] Present address: Department of Proteomics, The Novo Nordisk Foundation Center for Protein Research, Faculty of Health and Medical Sciences, University of Copenhagen, Copenhagen 2200, Denmark. [10] Present address: National Cerebral and Cardiovascular Center, 6-1 Kishibe-Shimmachi, Suita, Osaka 564-8565, Japan. [11] Present address: Laboratory of Laboratory/Sports Medicine, Division of Clinical Medicine, Faculty of Medicine, University of Tsukuba, 1-1-1 Tennodai, Tsukuba, Ibaraki 305-8577, Japan. ✉email: kanki@lsbm.org

*H*ox genes, which encode a family of homeodomain-containing transcription factors, are essential for the patterning of the anterior-to-posterior animal body axis during development. In mammals, the 39 *Hox* genes are divided into four clusters (*Hoxa*, *Hoxb*, *Hoxc*, and *Hoxd*) on four different chromosomes and consist of up to 13 paralogous groups. In each cluster, the *Hox* genes are arranged in tandem, from 3′ to 5′ (*Hox1* to *Hox13*). The 3′-paralogs are sequentially activated earlier than the 5′-paralogs along the anterior–posterior axis, a phenomenon that is called *Hox* temporal collinearity. This property of *Hox* expression confers special positional identities of the body segments, yet the underlying molecular mechanism is elusive. *Hox* transcription is switched on by retinoic acid signaling and morphogenic proteins, including Wnt and Fgf[1]. Once the transcription starts, the newly activated *Hox* gene loci progressively cluster into a transcriptionally active chromatin compartment[1–3]. Such transition in the spatial configuration coincides with the dynamics of chromatin histone marks, from a repressive state (tri-methylation of histone H3 lysine 27, H3K27me3) to a transcription-permissive state (tri-methylation of histone H3 lysine 4, H3K4me3)[4].

Polycomb group (PcG) proteins and the associated H3K27me3 mark maintain the state of transcriptional repression and gene silencing. Ezh2, a core component of the polycomb-repressive complex 2 (PRC2) is responsible for the methylation of H3K27me3. The *Hox* gene clusters are the best characterized PcG and H3K27me3 targets[5–7]. Indeed, mutation of the PcG genes induces ectopic *Hox* expression, resulting in a posterior transformation of the axial skeleton in mouse[8]. On the other hand, jumonji C (JmjC) domain-containing proteins, Utx and Jmjd3, specifically demethylate H3K27me2/3, and are involved in transcriptional activation of the *Hox* genes[9,10]. Although catalytic action of Utx has been implicated in the regulation of expression of the *Hox* genes during zebrafish development[9], it has been recently demonstrated that mouse with catalytically inactive Jmjd3, but not an Utx mutant, exhibits anterior homeotic transformation associated with a downregulation of *Hox* genes[11].

Di-methylation of histone H3 lysine 9 (H3K9me2), another repressive histone mark, is methylated by SET domain-containing proteins, G9a (encoded by *Ehmt2*) and GLP (encoded by *Ehmt1*)[12–14]. H3K9me2 is the most abundant heterochromatic histone modification, and covers large genomic domains in differentiated cells and in embryonic stem (ES) cells[15–17]. These domains are specifically associated with lamina-associated domains (LADs), characterized as transcriptionally repressive heterochromatin located within the nuclear peripheral region. A negative correlation between H3K9me2 deposition and gene expression is observed therein. During mouse embryogenesis, repressed *Hox* genes labeled by the H3K27me3 marks are located at a spatial domain distinct from the peripheral LADs[18]. Consistently, the association of genomic occupancies of H3K9me2 and H3K27me3 is mutually exclusive during the differentiation of the mouse ES cells[15,19]. Indeed, to the best of our knowledge, the functional relationship between *Hox* gene expression and the H3K9me2 histone mark has not been ruled out to date.

Another JmjC domain-containing protein, lysine demethylase 7A (Kdm7a), also known as Jhdm1d, contains a plant homeodomain (PHD), and is responsible for the demethylation of H3K9me2 and H3K27me2[20,21]. Kdm7a is predominantly expressed in mouse brain tissues[21]. Inhibition of a Kdm7a ortholog in zebrafish leads to developmental brain defects[21]. In mammalian neuronal cells, Kdm7a binds to the gene locus of *follistatin*, an antagonist of activin, which plays an important role in brain development. Kdm7a depletion suppresses the transcription of the gene, in association with increased levels of demethylated H3K9 and H3K27[21]. In addition, Kdm7a promotes neural differentiation of mouse ES cells by transcriptional activation of Fgf4, a signal molecule implicated in neural differentiation[20]. Knockdown of *Kdm7a* decreases Fgf4 expression, which correlates with the enriched coverage of both H3K9me2 and H3K27me2[20]. Furthermore, Kdm7a ortholog is predominantly expressed in epiblast cells of the primitive streak and promotes neural induction in an early chick embryo[22]. However, the biological role of Kdm7a during mouse development has not yet been reported.

Here, we report that $Kdm7a^{-/-}$ mutant mouse exhibits anterior homeotic transformation of the axial skeleton and downregulation of the transcription of posterior *Hox* genes during embryogenesis. Importantly, these changes in gene expression are associated with increased H3K9me2 but not H3K27me2 at the relevant posterior *Hox* loci. These observations demonstrate an essential role of Kdm7a on *Hox* gene regulation in vivo. Further, they provide evidence for the role of epigenetic histone mark H3K9me2 in the maintenance of *Hox* gene regulation during embryonic development in mouse.

## Results

### Construction of a $Kdm7a^{-/-}$ mouse by CRISPR/Cas9-mediated gene targeting

To disrupt the enzyme function of Kdm7a, we used a CRISPR/Cas9-based strategy to introduce a frameshift mutation at the start of the JmjC domain in *Kdm7a*. Because there are no suitable protospacer-adjacent motif (PAM) sequences in exon5 of the region encoding the JmjC domain, we designed single-guide RNAs (sgRNAs) located in exon6 of the region encoding the JmjC domain (Fig. 1a). To determine the optimal sgRNA sequence, we co-transfected HeLa cells with the pCAG-EGxxFP-target and pX330-sgRNA plasmids. We monitored the reconstituted enhanced green fluorescent protein (EGFP) fluorescence 48 h after transfection. Cetn1 was used as a positive control[23]. Although both sgRNA867 and sgRNA868 effectively cleaved the target site of pCAG-EGxxFP-Kdm7a, sgRNA868 worked slightly better than sgRNA867. We therefore selected sgRNA868 for further in vivo genome editing (Supplementary Fig. 1a).

Accordingly, we co-injected *Cas9* mRNA with sgRNA868 into pronuclear stage one-cell mouse embryos. The blastocysts derived from the injected embryos were then transplanted into foster mothers and newborn pups were obtained. Mice carrying the targeted mutations (chimera mice) were crossbred with the wild-type and heterozygous mice were obtained. Representative results of the restriction fragment length polymorphism (RFLP) analysis are shown in Supplementary Fig. 1b. We then amplified the Kdm7a-targeted regions by polymerase chain reaction (PCR), and subcloned and sequenced the PCR products, to confirm that the tested mice carried the mutant alleles with small deletions at the target site (Supplementary Fig. 1c). Since the phenotypes of mutant mice can differ between genetic backgrounds, especially for epigenetic factors[24,25], we generated *Kdm7a* mutant mice in both, ICR and C57BL/6 backgrounds and these mutant mice were used for subsequent analysis in vivo. Importantly, all mutant mice carried the frameshift mutations resulted in truncated Kdm7a proteins that lacked the core catalytic amino acid for its demethylase activity ($His^{284}$ at Fe(II)-binding site) (Fig. 1b)[21].

### Kdm7a regulates the anterior–posterior patterning of the axial skeleton in mouse

The *Kdm7a* mutant newborns appeared grossly normal. Considering that epigenetic factors, including histone demethylases, are associated with the anterior–posterior patterning[11,26–28], we investigated whether Kdm7a plays a role in the animal body patterning. To this end, we generated whole-mount skeletal preparations of postnatal day 1 wild-type and

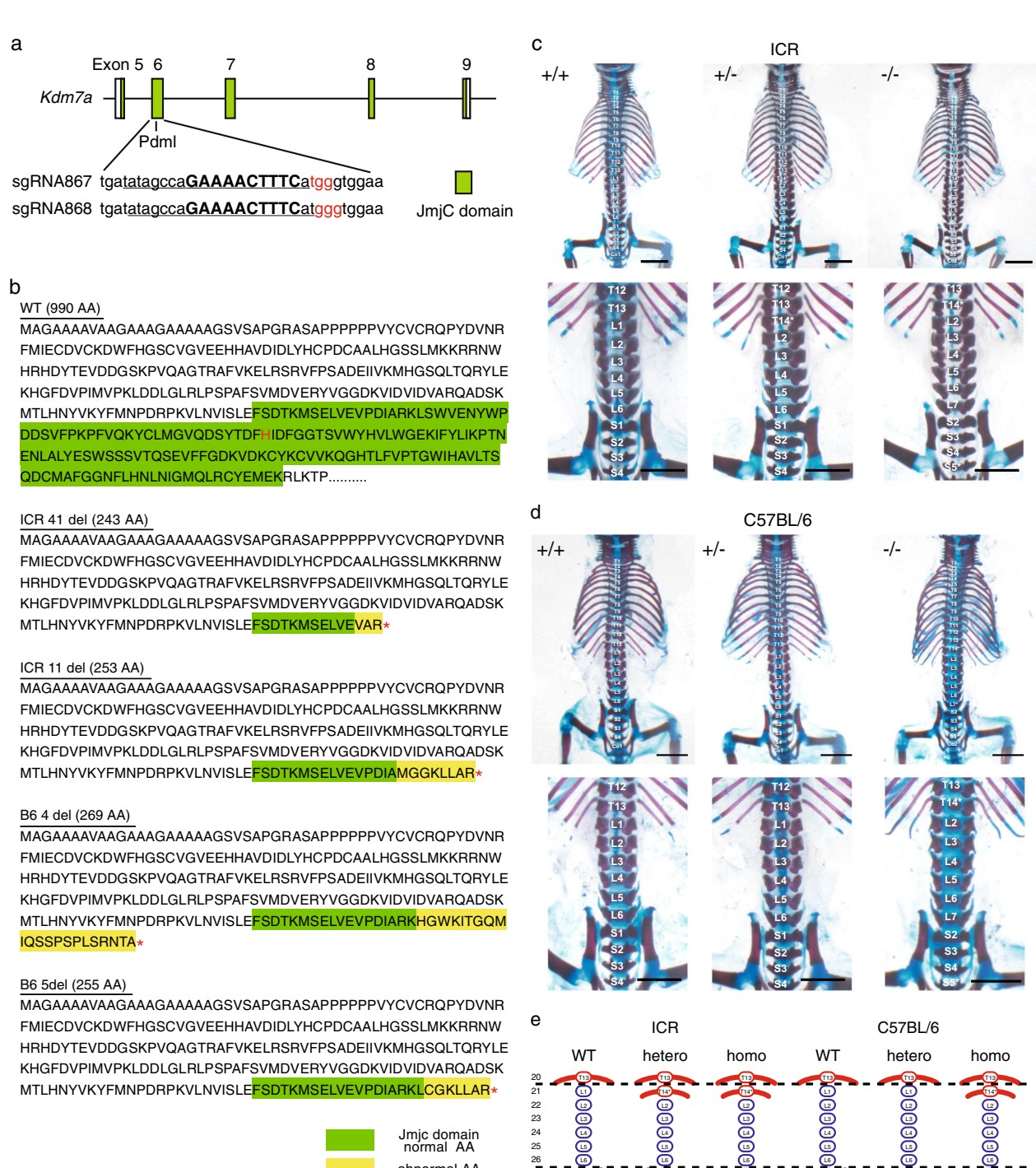

**Fig. 1 Kdm7a regulates the anterior–posterior patterning of the axial skeleton in mouse. a** Schematic of the Cas9/sgRNA-targeting sites in *Kdm7a*. The sgRNA-targeting sequence is underlined, and the protospacer-adjacent motif (PAM) sequence is labeled in red. The restriction sites in the target regions are bold and capitalized. Restriction enzymes used for restriction fragment length polymorphism (RFLP) are shown, and JmjC domain is shown as green boxes. **b** Amino acid (AA) sequence of *Kdm7a* KO mice from ICR or C57BL/6 backgrounds. All mutant mice carried frameshift mutation; the number of deleted nucleotides and total AA are shown. Normal and abnormal AA in the JmjC domain are highlighted with green and yellow, respectively. *Indicates termination of translation. His[284] at Fe(II)-binding site, a core catalytic AA for demethylase catalytic activity[21] is described in red. **c** Patterning defects in the axial skeleton of *Kdm7a* KO ICR background mouse. In *Kdm7a*[−/−] mice, the first lumbar (L1), the first sacral (S1) and the first coccygeal (Co1) vertebrae were transformed into thoracic (T14*), lumbar (L7*) and sacral (S5*) elements, respectively. In *Kdm7a*[+/−] mice, only L1 was transformed into thoracic (T14*) element. **d** Homeotic transformation in the axial skeleton of *Kdm7a* KO C57BL/6 mouse. In *Kdm7a*[−/−] mice, L1, S1 and Co1 were transformed into thoracic (T14*), lumbar (L7*) and sacral (S5*) elements, respectively. *Kdm7a*[+/−] background showed no patterning defects. **e** Summary of the patterning defects identified across *Kdm7a* mutant alleles in the ICR and C57BL/6 backgrounds. An asterisk indicates a homeotic transformation of the vertebral element.

**Table 1 Axial skeletal phenotypes of *Kdm7a* mutant mice (ICR background).**

|  | Wild-type | Hetero | Homo |
|---|---|---|---|
| Animal number | 5 | 5 | 11 |
| *Vertebral pattern* |  |  |  |
| T:L:S = 13:6:4 | 100% | 0% | 0% |
| (T1–T13, L1–L6, S1–S4) |  |  |  |
| T:L:S = 14:5:4 | 0% | 80% | 0% |
| (T1–T14*, L2–L6, S1–S4) |  |  |  |
| T:L:S = 14:6:4 | 0% | 20% | 100% |
| (T1–T14*, L2–L7*, S2–S5*) |  |  |  |

**Table 2 Axial skeletal phenotypes of *Kdm7a* mutant mice (C57BL/6 background).**

|  | Wild-type | Hetero | Homo |
|---|---|---|---|
| Animal number | 12 | 11 | 8 |
| *Vertebral pattern* |  |  |  |
| T:L:S = 13:6:4 | 100% | 100% | 0% |
| (T1–T13, L1–L6, S1–S4) |  |  |  |
| T:L:S = 14:5:4 | 0% | 0% | 0% |
| (T1–T14*, L2–L6, S1–S4) |  |  |  |
| T:L:S = 14:6:4 | 0% | 0% | 100% |
| (T1–T14*, L2–L7*, S2–S5*) |  |  |  |

*Kdm7a* mutant mice. As anticipated, all wild-type mice demonstrated the normal configuration of the axial skeleton, with 7 cervical, 13 thoracic, 6 lumbar and 4 sacral vertebrae (five out of five animals and 12 out of 12 animals from the ICR and C57BL/6 background, respectively) (Fig. 1c–e; Tables 1 and 2). By contrast, in all *Kdm7a*[−/−] mice displayed an anterior homeotic transformation of vertebral elements. The first lumber vertebra (L1) transformed into the thoracic element (T14*) gaining the ectopic ribs, and the 1[st] sacral (S1) and coccygeal (Co1) vertebrae showed transformation to lumbar (L7*) and sacral (S5*) identities with the loss and gain of connections to the pelvic girdle, respectively (all animals from both the ICR and C57BL/6 background) (Fig. 1c–e; Tables 1 and 2). Of note, even heterozygous mutant mice from ICR background exhibited an anteriorization of L1 into the thoracic element (T14*) (four out of five animals), while those from C57BL/6 background showed no difference from the normal vertebral disposition (all animals) (Fig. 1c–e; Tables 1 and 2), suggesting that the ICR genetic background has a much stronger influence on the anterior–posterior patterning in *Kdm7a* mutant mouse than the C57BL/6 background. Nonetheless, *Kdm7a* mutant mice showed the anterior homeotic transformation of the axial skeleton regardless of the genetic background (schematized in Fig. 1e). Considering that an ICR mouse is a non-inbred strain and genetically heterogeneous, we decided to conduct further detailed genetic analysis involving C57BL/6 mice.

**Kdm7a is involved in the regulation of *Hox* gene expression.** Loss-of-function of the murine *Hox* genes classically causes an anterior homeotic transformation[29]. Hence, we next examined the expression of *Hox* genes during embryogenesis by RNA sequencing (RNA-Seq). In the experiments, wild-type and *Kdm7a* mutant embryos at E9.5 and E10.5 were divided at the level of the otic vesicle (hereafter referred to as the "trunk") (beige-colored in Fig. 2a). The embryonic trunk at this developmental stage is a region in which the *Hox* genes are predominantly expressed[2]. Data for three biological replicates of RNA-Seq analyses were

highly correlated (Supplementary Fig. 2a). The analysis revealed that 73 and 2 genes were differentially expressed between the wild-type and *Kdm7a*[−/−] embryos at E9.5 and E10.5, respectively (padj <0.05; Fig. 2b; Supplementary Fig. 2b; Supplementary Data 1). A decreased expression of *Kdm7a* was detected in *Kdm7a*[−/−] embryo, which was probably associated with a nonsense-mediated mRNA decay[30]. Of note, many of the genes, including *Hox*, were downregulated in the *Kdm7a*[−/−] embryo, suggesting the possible role of Kdm7a in transcriptional activation (Fig. 2b; Supplementary Fig. 2b; Supplementary Data 1). This was consistent with a previous study showing that genetic ablation of H3K9me2 methyltransferase G9a resulted in the activation of many genes (upregulation of 147 and downregulation of 33 transcripts)[19]. As anticipated, gene ontology (GO) analysis and Ingenuity Pathway Analysis (IPA) of 73 differentially expressed genes, excluding *Kdm7a*, revealed a significant enrichment of the "skeletal system development", "anterior/posterior pattern specification", and "development of body axis" processes (Fig. 2c, d). In addition, the characteristics of these differentially expressed genes were related to the component "nucleus", the function "sequence-specific DNA binding", and the sequence domain "HOX" and "Homeobox, conserved site". This indicated that Kdm7a participates in the regulation of the developmental transcription factors, including Hox (Fig. 2c–e). The expression of other H3K9 and H3K27 histone methyltransferases and demethylases including Jmjd1a, G9a, Ezh2, and Utx was not altered (Supplementary Data 2), suggesting that the downregulation of Hox genes observed in *Kdm7a*[−/−] embryos may have not been caused by a secondary effect of transcriptional changes in other histone methyltransferases or demethylases.

Interestingly, when we focused on all (39) *Hox* genes, we observed that the posterior *Hox* genes were downregulated, while there were no differences in the expression in the anterior *Hox* genes in the *Kdm7a*[−/−] embryo compared with the wild-type (Fig. 2f). Quantitative PCR (qPCR) analysis confirmed that the expression of the majority of posterior *Hox* genes (*Hoxb6*; *c9*; *d9*; *a10*; *c10*; *d10*; *a11*; *c11*; *d11*; *d12*; and *d13* for E9.5; and *Hoxd8*; *c10*; *a11*; *c11*; *d12*; *a13*; *c13*; and *d13* for E10.5) was significantly decreased in the *Kdm7a*[−/−] embryo (Fig. 2g). By using whole-mount in situ hybridization, we examined the localization of Hoxd9 and Hoxd10 mRNA in embryos at E9.5. Although the transcript levels of Hoxd9 and Hoxd10 were decreased in the *Kdm7a*[−/−] embryos compared with wild-type, their spatial distribution was not altered (Fig. 2h), which was in line with a previous report showing that Jmjd3 regulates Hox gene expression levels but not its spatial distribution[11]. In addition, whole-mount in situ hybridization further revealed that, in wild-type embryos at E8.5, the expression of Kdm7a was observed in the primitive streak and presomitic mesoderm, where Hox9-10 genes were started to be activated during development[31] (Supplementary Fig. 3). This could be consistent with a previous report showing Kdm7a began to express in developmental head and tailbud of zebrafish at 24 post-fertilization, which is corresponding to E8.5 of mice[21]. Collectively, these findings support a functional role of Kdm7a-mediated transcriptional control, especially of the posterior *Hox* genes.

**H3K9me2 methylation is involved in the regulation of *Hox* genes.** Kdm7a-mediated demethylation of the repressive histone marks H3K9me2 and H3K27me2 correlates with active gene expression[20,21]. Therefore, we hypothesized that the transcriptional activation/repression of *Hox* genes during the anterior–posterior patterning would be associated with decreased/increased levels of H3K9me2 and H3K27me2. To test this hypothesis, we characterized the epigenetic landscape in the Hox-

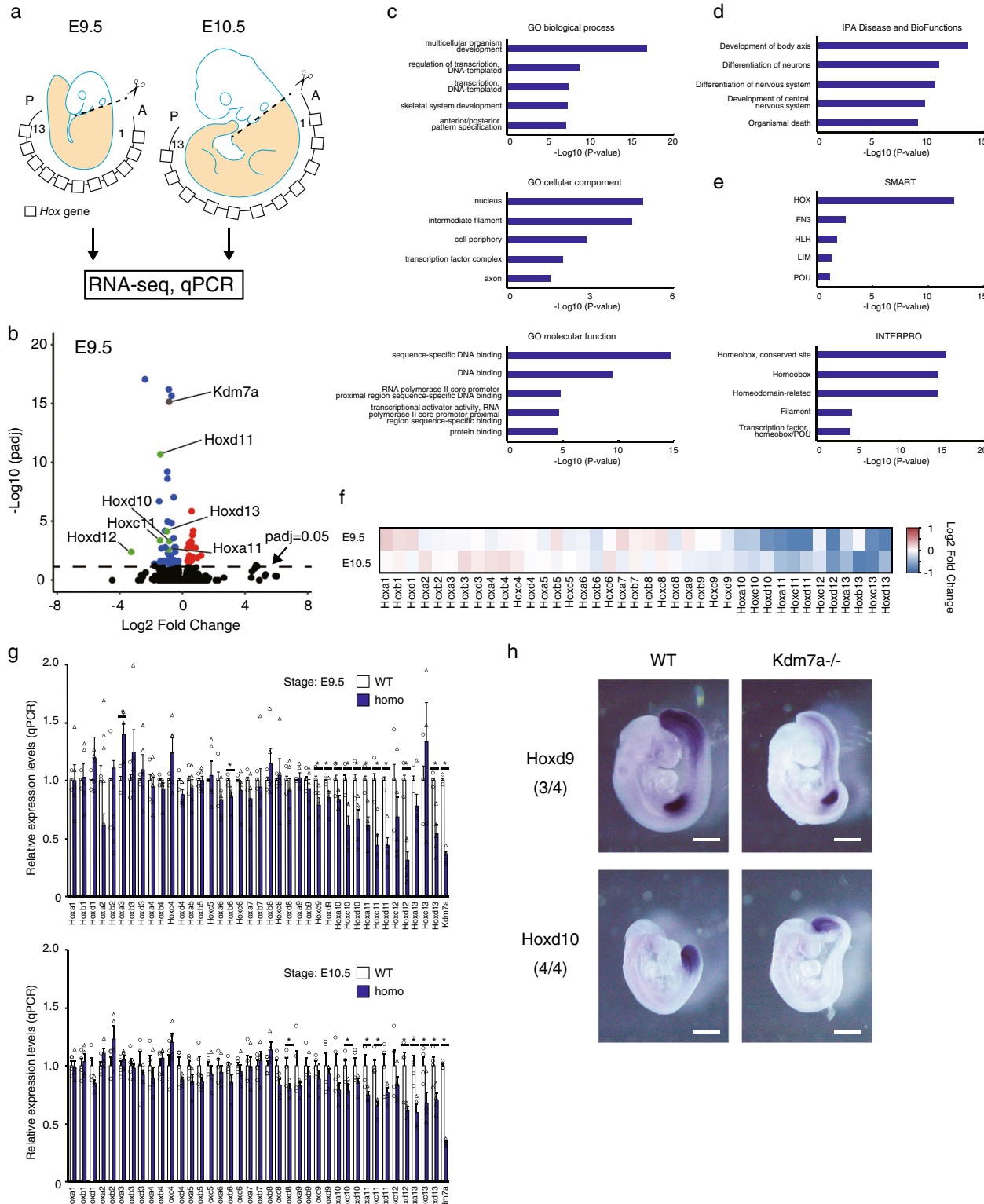

inactive developmental brain[32] (hereafter referred to as the "head") versus Hox-active developmental trunk (Supplementary Fig. 5a). Although the expression of *Hox* genes is more subject to change at E9.5 than E10.5, we selected the latter time point due to the requirement for a large number of cells for the chromatin immunoprecipitation (ChIP) analysis. Biological replicates of ChIP-Seq analyses were highly correlated (Supplementary Fig. 4a–d). Consistent with a previous report[4], we observed, in the developmental trunk where Hox genes are actively transcribed,

the entire deposition of an active histone mark, H3K4me3, at a representative *Hoxa* cluster that was paralleled by relatively low enrichment of repressive histone mark, H3K27me3. Conversely, H3K27me3 covered the entire *Hoxa* gene cluster in the developmental head, where Hox genes are inactive, which was associated with extremely low enrichment of H3K4me3 (Supplementary Fig. 5b, c). Importantly, we observed a higher enrichment of H3K9me2 in the head compared to the trunk at the representative *Hoxa* cluster, while these for H3K27me2 were

**Fig. 2 Kdm7a is involved in the regulation of *Hox* gene expression. a** Schematic of E9.5 and E10.5 mouse embryo microdissection. *Hox* genes are located in tandem along a chromosomal locus, and are sequentially activated along the anterior–posterior axis during embryogenesis. The posterior part of the embryo (beige; referred to as the "trunk") was used for RNA sequencing (RNA-Seq) and quantitative polymerase chain reaction (qPCR) analysis. A and P indicate anterior and posterior, respectively. The described numbers indicate the number of *Hox* genes. **b** Volcano plots showing differentially expressed genes in the wild-type and $Kdm7a^{-/-}$ embryos ($n = 3$ for each genotype) at E9.5 The $X$- and $Y$-axes indicate the log2 fold-change and –log10 adjusted $P$-value (padj) produced by DESeq2, respectively. Genes with padj < 0.05 are indicated as red (increase) and blue (decrease) spots. **c–e** Gene ontology (GO) analysis (**c**), Ingenuity Pathway Analysis (IPA) (**d**), and domain prediction analysis (InterPro and SMRT) (**e**) of 73 differentially expressed genes in $Kdm7a^{-/-}$ mouse, as determined in (**b**). The $P$-values for each category are shown in the bar graphs. **f** Heatmaps showing the log2 fold-change expression differences (determined by DESeq2) in *Hox* genes between the wild-type and $Kdm7a^{-/-}$ embryos at E9.5 and E10.5. Red to blue coloring indicates the fold-change. (**g**) qPCR analysis comparing the expression of *Hox* genes between wild-type and $Kdm7a^{-/-}$ embryos at E9.5 (top; $n = 5$ for each genotype) and E10.5 (bottom; $n = 4$ for each genotype). The average number of somites in the wild-type and $Kdm7a^{-/-}$ was 26 and 24 at E9.5, respectively, and 39 and 42 at E10.5, respectively (there were no statistically significant differences between the wild-type and $Kdm7a^{-/-}$ embryos). Data are shown as means ± SE. *$P < 0.05$ compared with the wild-type. Statistical differences were analyzed by the Student's $t$ test. **h** Whole-mount in situ hybridization of *Hoxd9* (top) and *Hoxd10* (bottom) mRNA in the wild-type (left) and $Kdm7a^{-/-}$ (right) embryos at E9.5. The numbers of $Kdm7a^{-/-}$ embryos presenting decreased levels of Hox gene expression are indicated.

not altered. For more quantitative comparison, ChIP-Seq signals were normalized using input libraries, and average heatmap of input-normalized ChIP-Seq signals for H3K9me2 further revealed higher enrichment of H3K9me2 at almost all *Hox* genes in the head regions compared to the trunk (Supplementary Fig. 5d). Furthermore, ChIP followed by qPCR confirmed an increase in H3K9me2 and H3K27me3 levels and a decrease in H3K4me3 levels in the vicinity of the transcription start site (TSS) of *Hoxa3* and *Hoxa13*, but no changes at the *actin beta* (*Actb*) site (Supplementary Fig. 5e). Nevertheless, we detected no differences in H3K27me2 between the head and trunk (Supplemental Fig. 5e), suggesting that Kdm7a might not regulate Hox genes expression through H3K27me2-mediated mechanisms.

We next examined whether ablation of Kdm7a affected the epigenetic landscape at the *Hox* genes in the developmental trunk regions (Fig. 3a). ChIP-Seq analysis demonstrated relatively high occupancy of H3K9me2 at the representative *Hoxa* locus in the $Kdm7a^{-/-}$ embryonic trunk compared to wild-type, but no differences in the levels of H3K27me2 (Fig. 3b, c). Despite the fact that opposed labeling of H3K4me3 and H3K27me3 is involved in the regulation of Hox genes during development[4], there were no obvious differences in the levels of H3K4me3 and H3K27me3 between the wild-type and $Kdm7a^{-/-}$ embryonic trunk (Fig. 3b, c). In accordance with the mRNA expression data (Fig. 2f, g), average heatmap of input-normalized ChIP-Seq signals for H3K9me2 further revealed that H3K9me2 coverage was moderately enriched at the posterior *Hox* genes in the $Kdm7a^{-/-}$ embryo in comparison with the wild-type (Fig. 3d). Consistently, ChIP followed by qPCR showed an increase in H3K9me2 levels in the vicinity of the TSS of *Hoxa3* and *Hoxa13*, but no changes at *Actb* (Fig. 3e). We repeatedly detected no differences in H3K4me3, H3K27me2, and H3K27me3 levels between the wild-type and the $Kdm7a^{-/-}$ trunk at *Actb*, *Hoxa3*, and *Hoxa13* loci in ChIP followed by qPCR analysis (Fig. 3e). Taken together, these observations suggest the possibility that Kdm7a-mediated regulation of the repressive histone mark H3K9me2 might be involved in transcriptional activation of the *Hox* genes.

## Discussion
Histone-modifying enzymes have been recognized as key players during early development and differentiation, as well as various diseases. Kdm7a, a histone demethylase for H3K9me2 and H3K27me2, is reportedly involved in neural differentiation of mouse ES cells and in brain development in zebrafish[20,21]. In addition, Kdm7a is highly induced in cancer cells in response to nutrient starvation and is associated with tumor suppression, by modulating tumor angiogenesis[33]. However, its role in mouse development has not been elucidated. We report here the generation and characterization of a previously undescribed *Kdm7a* mouse mutant. We provide the evidence that Kdm7a is involved in the activation of the posterior *Hox* gene expression, and subsequent patterning of the anterior–posterior body axis, in vivo. Since we observed increased levels of H3K9me2 but not H3K27me2 at the relevant posterior *Hox* loci in the *Kdm7a* mutant embryo, we propose that Kdm7a modulates the developmental *Hox* gene activation by regulating the repressive histone mark H3K9me2 (Fig. 4).

Many studies support the notion that the mammalian *Hox* genes are targets of PcG proteins and their associated H3K27me3, indicating the essential role of H3K27me3 in the silencing of *Hox* gene expression[4,18,34,35]. Consistently, our results confirmed the opposed labeling of H3K27me3 between Hox-inactive developmental head and Hox-active developmental trunk. In the mouse ES cells, H3K27me3 covers the entire *Hox* clusters, in which the *Hox* genes are transcriptionally repressed. Further, during collinear activation of the *Hox* genes, H3K27me3 marks at the *Hox* cluster loci are progressively diminished in the sequence of transcriptional activation of the *Hox* genes[4]. Accordingly, posterior transformation and an increased expression of the *Hox* genes are commonly seen in mice lacking PcG proteins[8]. For example, a mouse with mutation in the *Mel18* gene, also known as the PcG ring finger 2 gene, exhibits a posteriorizing shift of body axis (e.g., the loss of rib in the thoracic vertebrae and ectopic ribs in the cervical vertebrae)[36,37]. Furthermore, *Jmjd3* mutant mouse, in which the protein's H3K27 demethylation domain is disrupted, exhibits an anterior homeotic transformation (e.g., the gain of rib in the lumbar vertebra), which is associated with the downregulation of *Hox* genes[11]. No involvement of H3K27me3 was observed in the deregulation of *Hox* genes and anterior transformation in $Kdm7a^{-/-}$ embryos, suggesting the possible mechanisms for H3K27me3 independent transcriptional repression of Hox genes, but further detailed studies are warranted to confirm these findings.

While both H3K9me2 and H3K27me3 are involved in facultative heterochromatinization during the development, limited overlap between H3K9me2 and H3K27me3 targets has been suggested[38]. Indeed, during the early postimplantation development, only a few genes are differentially expressed between the $Ehmt2^{-/-}$ and $Ezh2^{-/-}$ mutant embryos, which is in line with H3K9me2 and H3K27me3 being linked to distinct repressive chromatin states[19]. Accordingly, genome-wide analysis of the differentiation of mouse ES cells revealed that the occurrence of H3K9me2 and H3K27me3 is mutually exclusive, with relatively sharp boundaries between the two marks[15]. Supportively, our results demonstrated, in $Kdm7a^{-/-}$ embryos, an increased level of H3K9me2 at *Hox* gene loci without any changes of H3K27me3.

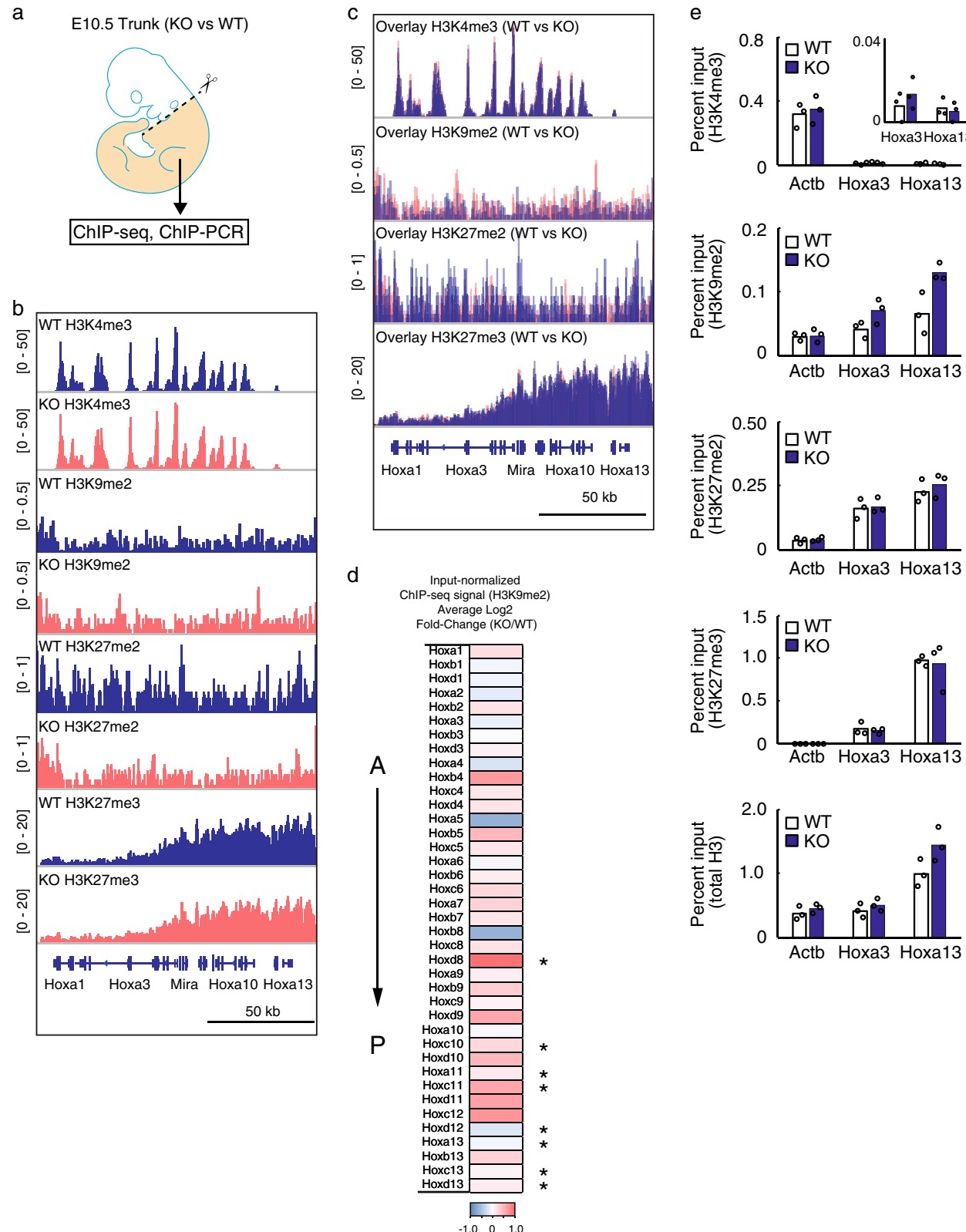

Another example concerns "germline genes", which are crucial for the progression of the primordial germ cell to meiosis in the female and for transposon repression in the male. These genes are silenced by both H3K9me2 and H3K27me3 in the mouse ES cells, and are progressively activated in association with decreased H3K9me2 marks, along with the specification and development of the primordial germ cell[39]. In addition, deposition of both

H3K9me2 and H3K27me3 in the vicinity of the genomic region of *Pax5*, regulated by PcG proteins, is simultaneously decreased by *Ehmt2* knockout in the mouse ES cells[16]. Furthermore, siRNA knockdown of *Kdm7a* in neuronal cells led to increased levels of not only H3K9me2 but also H3K27me3 at the *follistatin* locus[21]. Taken together, H3K9me2 and H3K27me3 normally have independent functions, as observed in mouse ES cells. Nevertheless,

**Fig. 3 H3K9me2 methylation is involved in the regulation of *Hox* genes. a** Developmental trunks from the wild-type or *Kdm7a*⁻/⁻ embryos at E10.5 (beige) were used for chromatin immunoprecipitation (ChIP)-Seq and ChIP-qPCR. **b, c** Gene tracks of ChIP-Seq signals for H3K4me3, H3K9me2, H3K27me2, and H3K27me3 close to the *Hoxa* cluster in the trunk-region of the wild-type and *Kdm7a*⁻/⁻ embryos. ChIP-Seq signals were visualized using Integrative Genomics Viewer (http://software.broadinstitute.org/software/igv/) on the separate (**b**) and overlay (**c**) view. **d** Heatmaps showing the average log2 fold-change of input-normalized H3K9me2 ChIP-Seq signals in the *Hox* genes between the trunk regions from the wild-type and *Kdm7a*⁻/⁻ embryos. Red to blue coloring indicates the fold-change. A and P indicate anterior and posterior, respectively. *indicates Hox genes that were significantly downregulated in E10.5 Kdm7a⁻/⁻ embryos determined by qPCR (Fig. 2g). **e** ChIP-qPCR of H3K4me3, H3K9me2, H3K27me2, H3K27me3, and total H3 at the *Actb*, *Hoxa3*, and *Hoxa13* TSS in the trunk regions of the wild-type and *Kdm7a*⁻/⁻ embryos, normalized to input. Graphs are representative of two or three independent experiments. The data represent means from $n = 3$ technical replicates; independent experiments were repeated two or three times with similar results.

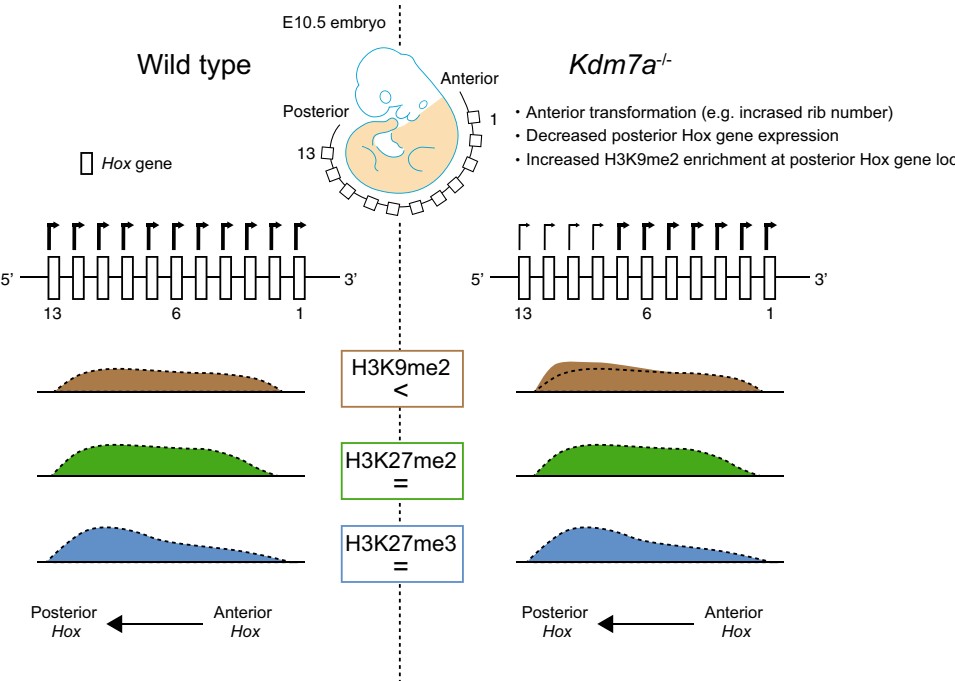

**Fig. 4 A schematic model of epigenetic regulation by Kdm7a during developmental *Hox* genes activation.** Posterior Hox genes are specifically downregulated in Kdm7a⁻/⁻ mice in comparison with wild-type mice, which is associated with increased enrichment of H3K9me2 but not H3K27me2 and H3K27me3.

cooperative transcriptional control by H3K9me2 and H3K27me3 could occur under certain conditions.

One of the two classes of PcG, PRC2, has methylation capacity for not only H3K27me3 but also H3K27me2[40]. Unlike H3K27me3, H3K27me2 is widely abundant, marking 50%–70% of total histone H3 and covering inter- and intragenic regions, suggesting that its role is to prevent inappropriate promoter or enhancer activities[40–42]. To date, H3K27me2 is considered as an intermediate H3K27 methylation state that marks genes as being potentially activated or inactivated. Regarding transcriptional control of Hox genes, the role of H3K27me2 is controversial and thus remains be fully elucidated. For example, knock down of PHF1, a core component of PRC2, caused upregulation of Hox genes via redistribution of Ezh2, which was linked with decreased H3K27me2 and H3K27me3 levels in NIH3T3 cells[43], while also linked with increased H3K27me2 and decreased H3K27me3 in HeLa cells[44]. Kdm7a reportedly can demethylate H3K27me2, suggesting the possibility that Kdm7a could transcriptionally regulate Hox gene expression through H3K27me2 dependent mechanism[20,21]. Importantly, H3K27me2 coverage was not altered between the wild-type and *Kdm7a*⁻/⁻ developmental trunk, or between the head, where Hox genes are inactive, and the trunk, where Hox genes are active. Thus, in mice, at least during the developmental period that we observed in this study, H3K27me2 might not play a

dominant role in the transcriptional regulation of Hox genes, as well as anterior–posterior axial development.

Recent findings have suggested a non-catalytic function of histone-modifying enzymes, especially in tumorigenesis. UTX-mediated chromatin remodeling suppresses acute myeloid leukemia via a noncatalytic inverse regulation of the oncogenic and tumor-suppressive transcription factor programs[45]. In addition, a non-enzymatic function of SETD1A, a methyltransferase of H3K4, regulates the expression of genes involved in DNA damage response and is required for the survival of acute myeloid leukemia cells[46]. In the present study, we showed that H3K9me2 occupancy is enriched in the *Hox*-negative developmental brain and increased in the posterior part of the *Kdm7a* mutant embryo. Hence, we believe that the catalytic activity of Kdm7a possibly plays an important role in the transcriptional control of *Hox* genes during embryogenesis. Nonetheless, experiments involving a catalytically inactive mutant will be required to clarify this point.

In conclusion, the presented data establish an important in vivo role of Kdm7a in the anterior–posterior axial development. Kdm7a regulates the transcription of *Hox* genes most likely by the demethylation of the repressive histone mark H3K9me2. Such systems might be essential for the proper control of coordinate body patterning in vertebrate development. Currently,

studies focusing on the role of H3K9me2 during embryogenesis are limited[47], and further studies are warranted to understand the mechanisms through which H3K9me2 mediates transcriptional regulation of the developmental genes, including *Hox*.

## Methods

**Mice**. All mouse experiments were approved by The University of Tokyo Animal Care and Use Committee (approval number H28-1). The animals were housed in individual cages in a temperature- and light-controlled environment, and had ad libitum access to chow and water. All mouse experiments were approved by The University of Tokyo Animal Care and Use Committee.

**Cell lines**. Human cervical cancer cell line, HeLa, was purchased from ATCC (Manassas, VA) and grown and passaged every 2 or 3 days in DMEM (nacalai tesque, Kyoto, Japan), supplemented with 1% penicillin/streptomycin (Wako, Osaka, Japan) and 10% FBS (Thermo Fisher Scientific, Waltham, MA). The cells were cultured at 37 °C and in a 5% $CO_2$ atmosphere in a humidified incubator.

**Plasmids and mRNA preparation**. The pCAG-EGxxFP[23] plasmid was a kind gift from Dr. M Ikawa (The University of Osaka). Genomic fragments (~500-bp) containing the sgRNA target sequence were PCR-amplified and placed between the EGFP-encoding fragments. Plasmids expressing both hCas9 and sgRNA were prepared by inserting synthetic oligonucleotides (Hokkaido System Science, Hokkaido, Japan) at the BbsI site of pX330 (http://www.addgene.org/42230/)[48]. Plasmids pCAG-EGxxFP, harboring the sgRNA target sequence of *Cetn1*, and pX330, containing sgRNA-targeting *Cetn1*, were also kindly gifted from Dr. M Ikawa[23]. The p3s-Cas9HC plasmid (https://www.addgene.org/43945/) was used to generate hCas9 mRNA. The plasmid for producing sgRNA was prepared by inserting synthetic oligonucleotides (Hokkaido System Science) at the BsaI site of DR274 (https://www.addgene.org/42250/). RNA was synthesized from the XbaI-digested p3s-Cas9HC plasmid by using mMESSAGE mMACHINE T7 ULTRA transcription kit (Thermo Fisher Scientific) in accordance with manufacturer's protocol. RNA was synthesized from the DraI-digested DR274 plasmid by using MEGAshortscript™ T7 transcription kit (Thermo Fisher Scientific) in accordance with manufacturer's protocol. The hCas9 mRNA and sgRNAs were purified by phenol chloroform-isoamyl alcohol extraction and isopropanol precipitation, followed by spin column chromatography using NANOSEP MF 0.2 μm (Thermo Fisher Scientific). The PCR primers and oligonucleotide sequences for the constructs are listed in Supplementary Table 1.

**Transfection procedure**. For the experiment, 250 ng of pCAG-EGxxFP-target was mixed with 250 ng of pX330 harboring the sgRNA sequences, and the mixture was used to transfect $1 \times 10^5$ HeLa cells in a well of a 24-well plate using the Lipofectamine® LTX reagent (Thermo Fisher Scientific), according to the manufacturer's protocol. The EGFP fluorescence was observed under a confocal microscope (C2⁺ Confocal Microscope System; Nikon, Tokyo, Japan) 48 h after the transfection.

**Pronuclear injection**. ICR and C57BL/6 female mice were superovulated and mated with ICR and C57BL/6 males, respectively, and fertilized eggs were collected from the oviduct. Then, the hCas9 mRNA (0.05 μg/μl) and sgRNAs (0.05 μg/μl) were co-injected into pronuclear-stage eggs. The eggs were cultivated in kSOM overnight and then transferred into the oviducts of pseudopregnant ICR females.

**Genotyping**. Mouse genomic DNA samples were prepared from tail biopsies. PCR was performed using *Kdm7a*-specific primers to amplify the sgRNA target site (Supplementary Table 1), and under the following cycling conditions: 95 °C for 10 min; followed by 40 cycles of 95 °C for 20 s, 60 °C for 20 s, and 72 °C for 30 s; incubation step at 72 °C for 7 min; and hold at 4 °C. BMS BIOTAQ™ DNA polymerase (Nippon Genetics Co. Ltd, Tokyo, Japan) was used for PCR reactions. The *Kdm7a* PCR product was digested with XmnI (New England Biolabs, Beverly, MA). The digested DNA was resolved on an ethidium bromide-stained agarose gel (2%). For sequencing, PCR products were cloned using the DynaExpress TA PCR cloning kit (BioDynamics Laboratory Inc, Tokyo, Japan), and the mutations were identified by Sanger sequencing.

**Skeletal staining**. Alizarin red and alcian blue staining were performed, as previously described[49]. Samples (postnatal day 1 mice) were fixed in 95% ethanol for 1 week, placed in acetone for 2 days, and then incubated with 0.015% alcian blue 8GS, 0.005% alizarin red S, and 5% acetic acid in 70% ethanol for 3 days. After washing in distilled water, the samples were cleared in 1% KOH for at least 2 days and then in 1% KOH glycerol series until the surrounding tissues turned transparent. The specimens were stored in glycerol until morphological analysis under a stereomicroscope.

**Dissection of the anterior and posterior parts of the embryo**. Dissection of the anterior and posterior parts of the embryo (referred to as the "head" and "trunk", respectively) was performed as described previously[27,50], with minor modifications. In brief, the wild-type and *Kdm7a⁻/⁻* embryos were staged precisely by counting the somites. Embryos at somite stage 25 (E9.5) and 40 (E10.5) were used for the majority of experiments in the current study. For genomic and transcriptomic analysis, embryos were dissected from *Kdm7a⁻/⁻* mice and their respective littermate control mice. The embryonic head and trunk were divided at the level of otic vesicle, by utilizing micro-surgical scissors. The embryonic head and trunk were then transferred directly to QIAzol® lysis reagent (Qiagen, Hilden, Germany) and were stored at –80 °C for RNA isolation.

**mRNA isolation**. Total RNA was isolated from the embryonic head and trunk by using a miRNeasy micro kit (Qiagen) with the DNase digestion step, according to the manufacturer's instructions.

**qPCR for mRNA quantification**. The isolated RNA (500 ng) was reverse-transcribed to cDNA by using PrimeScript RT master mix (Takara, Shiga, Japan). PCR was performed using a CFX96 unit (Bio-Rad, Hercules, CA) with SYBR® Premix EX Taq™ II (Takara). The relative expression levels were calculated using *β-actin* mRNA as a reference. The primers used for these analyses are listed in Supplementary Table 2.

**Whole-mount in situ hybridization**. Whole-mount in situ hybridization was performed as described previously[51]. Probes for *Hoxd9* and *Hoxd10* were kindly gifted by Dr. H. Hamada (The University of Osaka). Probes for *Kdm7a* was obtained by RT-PCR using the forward primer 5′- GAGTCTTCCCAAGTGCC-GATGA-3′ and the reverse primer 5′- AGAACACCTCACTCTGGGTCAC-3′.

**ChIP-qPCR**. The embryonic head and trunk were collected as described in the section *Dissection of the anterior and posterior parts of the embryo*. To prepare single-cell suspension, the tissues were placed in 1 ml of phosphate-buffered saline, pipetted and passed through a 35-μm cell strainer (Corning Japan, Tokyo, Japan). The cells were fixed for 10 min in a 1% formaldehyde solution at room temperature and then neutralized for 5 min in a 0.125 M glycine solution. Pooled tissue samples from two embryos were used in ChIP analysis. ChIP was performed as previously described[52,53]. Briefly, fixed cells were re-suspended in 2 ml of sodium dodecyl sulfate lysis buffer, containing 10 mM Tris-HCl, pH 8.0 (Thermo Fisher Scientific), 150 mM NaCl (Thermo Fisher Scientific), 1% sodium dodecyl sulfate (Sigma-Aldrich, St. Louis, MO), 1 mM EDTA, pH 8.0 (Thermo Fisher Scientific), and cOmplete™ EDTA-free protease inhibitor cocktail (Sigma-Aldrich). The samples were then fragmented in a Picoruptor (40 cycles, 30 s on/30 s off; Diagenode, Liege Science Park, Belgium). The sonicated solution was diluted with ChIP dilution buffer [20 mM Tris-HCl, pH 8.0, 150 mM NaCl, 1 mM EDTA, and 1% Triton X-100 (Sigma-Aldrich)] up to 10.3 ml; 10 ml were used for immunoprecipitation (10 ml) and the remaining 300 μl were saved as non-immunoprecipitated chromatin (the input sample). Specific antibodies against H3K4me3, H3K9me2, and H3K27me3 (MAB Institute, Inc. Nagano, Japan), and H3K27me2 (Cell Signaling Technology, Danvers, MA), and total H3 (Abcam, Cambridge, MA) were bound to magnetic Dynabeads M-280 (Thermo Fisher Scientific) and mixed with the diluted, sonicated solution for immunoprecipitation. The prepared DNA was quantified using a NanoDrop 2000 spectrophotometer (Thermo Fisher Scientific), and more than 10 ng of DNA were processed for qPCR. The quantification primers are listed in Supplementary Table 3. PCR was performed using a CFX96 PCR and SYBR® Premix EX Taq™ II. Fold enrichment was determined as the percentage of the input.

**ChIP-Seq library preparation**. ChIP-Seq library was prepared using DNA sonicated to an average size of 0.5 kb. ChIP samples were processed for library preparation using a KAPA Hyper Prep kit (Kapa Biosystems Inc., Wilmington, MA), according to the manufacturer's instructions. Deep sequencing was performed using a HiSeq 2500 sequencer (Illumina Inc., San Diego, CA) as single-end 36-b reads.

**RNA-Seq library preparation**. Total RNA from the embryos was isolated as described above in the section *mRNA isolation*. The RNA integrity score was calculated using the RNA 6000 Nano reagent (Agilent Technologies) and a 2100 Bioanalyzer (Agilent Technologies). RNA integrity value (RIN) score of all samples used for the preparation of RNA-Seq libraries was above 9. RNA-Seq libraries were prepared with a TruSeq RNA Library Prep Kit (Illumina). The libraries were sequenced using a HiSeq 2500 sequencer (Illumina) as paired-end 150-b reads.

**Bioinformatics**

*RNA-Seq data analysis*. The quality of FASTQ files was checked by using FastQC (http://www.bioinformatics.babraham. ac.uk/projects/fastqc) version 0.11.8, and trimmed using Trimmomatic PE version 0.38[54] with "ILLUMINACLIP:adaptor_sequence.fa:2:30:7:1:true LEADING:3 TRAILING:3 SLIDINGWINDOW:4:15 CROP:120 MINLEN:36" parameters. The trimmed FASTQ files were aligned to the

mouse reference genome mm10 using Hisat2 version 2.1.0[55] with a "–dta" option. SAM files were sorted and converted into BAM files using Samtools version 1.9[56]. Gene expression was quantified in transcripts per kilobase million (TPM) using StringTie version 1.3.4d[57] with an "-e" parameter; the GTF file was downloaded from GENCODE release M20 (https://www.gencodegenes.org/mouse/release_M20.html) and input with a "-G" option. To visualize the sequencing tracks, BIGWIG files were generated from BAM files using deepTools version 3.2.0[58], bamCoverage command with "-of bigwig -bs 1–exactScaling–normalizeUsing CPM" parameters, and displayed in Integrative Genomics Viewer[59]. Read count table was produced using feature-Counts version 1.6.3 with "-t exon -g gene_id–extraAttributes gene_name -M -s 0 -p -P -d 0 -D 500 -a gencode.vM20.annotation.gtf" parameters. Differential expression was determined using DESeq2[60] by testing wild-type versus $Kdm7a^{-/-}$ embryos at E9.5 or E10.5. The values obtained from DESeq2 were used to generate a heatmap and volcano plots. Differentially expressed genes were defined based on two criteria: (1) padj < 0.05 and (2) TPM > 1 in either or both wild-type or KO samples, and used for GO analysis in DAVID[61] and IPA (QIAGEN, https://www.qiagenbioinformatics.com/products/ingenuity-pathway-analysis).

*ChIP-Seq data analysis*. The quality of FASTQ files was by using FastQC (http://www.bioinformatics.babraham. ac.uk/projects/fastqc) version 0.11.8, and trimmed using Trimmomatic SE version 0.38[54] with "ILLUMINACLIP:adaptor_sequence.fa:2:30:7 LEADING:3 TRAILING:3 SLIDINGWINDOW:4:15 MINLEN:36" parameters. The trimmed FASTQ files were aligned to the mouse reference genome mm10 using Bowtie2 version 2.3.4.3[62] with a "-N 1" option. SAM files were sorted and converted into BAM files using Samtools version 1.9[56]. To visualize the sequencing tracks, BIG-WIG files were generated from BAM files using deepTools version 3.2.0[58] bamCoverage command with "-of bigwig -bs 50–exactScaling–normalizeUsing CPM -e 500" parameters, and displayed in Integrative Genomics Viewer[59]. To generate the signal heatmaps (Fig. 3d; Supplementary Fig. 5d), log2 fold-change of the input-normalized ChIP-Seq signals for each *Hox* gene were calculated as follows:

$$\log_2\left\{\left(CPM^{IP1}/CPM^{input1}\right)/\left(CPM^{IP2}/CPM^{input2}\right)\right\},$$

where $CPM^{IP}$ and $CPM^{input}$ are the read counts per million mapped reads (CPM) from IP and input libraries, respectively. $CPM^{IP}$ was calculated over gene bodies, while $CPM^{input}$, a local background, was calculated over *Hox* cluster loci to minimize the effect of site-specific noises. *Hox* cluster loci were defined as intervals from the first gene (*Hox1* or *Hox4*) to the last gene (*Hox13*); namely, as follows: *Hoxa* cluster, chr6:52155590-52260880; *Hoxb* cluster, chr11:96194316-96368256; *Hoxc* cluster, chr15:102921103-103036852; *Hoxd* cluster, chr2:74668310-74765142. Reads were counted using the feature Counts version 1.6.3 with "–readExtension3 500 -M -O -s 0" parameters, in which a custom GTF file was prepared for counting reads in *Hox* cluster loci. CPM normalization was performed using a custom R script. Finally, average log2 fold-change of two biological replicates was shown as heatmaps.

**Statistics and reproducibility**. In each biological experiment, at least two or three independent repeats were performed. RNA-Seq and ChIP-Seq experiments were done with three and two biological replicates, respectively, and each reproducibility was confirmed by correlation coefficients. Statistical differences were analyzed by using the Student's *t*-test. In all tests, differences at *P*-values of <0.05 were considered to be statistically significant.

**Reporting summary**. Further information on research design is available in the Nature Research Reporting Summary linked to this article.

## Data availability

Sequence data can be accessed through the Gene Expression Omnibus (GEO) under the NCBI accession number GSE133189. The summary of RNA-Seq analysis is shown as Supplementary Data 1. RNA-Seq analysis for histone methyltransferases and demethylases is shown as Supplementary Data 2. The source data underlying the graphs presented in the figures are shown as Supplementary Data 3.

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

## Acknowledgements

We thank Shiro Fukuda and Shogo Yamamoto (The University of Tokyo) for bioinformatics analysis. pX330 plasmid vectors were kindly gifted from Dr. Masahito Ikawa (Osaka University). This work was supported by a Grant-in-Aid for JSPS Postdoctoral Fellows (to Y.H.); a Grant-in-Aid for Young Scientists (B) 17K15991 (to Y.H.); a Grant-in-Aid for Young Scientists (A) 26710013 (to Y. Kanki); a Grant-in-Aid for Scientific Research on Innovative Areas (Research in a Proposed Research Area) 25125707 (to Y. Kanki); a Grant-in-Aid for Challenging Exploratory Research [26670397 (to Y. Kanki) and 16K15438 (to Y. Kanki)]; a Fund for the Promotion of Joint International Research (Fostering Joint International Research) 15KK0251 (to Y. Kanki); a Research Grant from Nanken-Kyoten, TMDU (to Y.H., Y. Kanki, Y.W., and T.F.); a Research Grant from Takeda Science Foundation (to Y. Kanki); a Research Grant from the Japan Heart Foundation (to Y. Kanki); a Research Grant from MSD Life Science Foundation (to Y. Kanki); a Research Grant from Uehara Memorial Foundation (to Y. Kanki); a Research Grant from SENSHIN Medical Research Foundation (to Y. Kanki); and a Research Grant from Kowa Life science Foundation (to Y. Kanki).

## Author contributions

Y.H., T.K., Y. Kawamura, and Y. Kanki designed the research strategies; Y.H., M.Y., T.K., Y. Kawamura, A.T., N. Nakada, and Y. Kanki performed the experiments; Y.H., N. Nagai, and Y. Kanki performed the bioinformatic analyses; Y.H., N. Nagai. T.K., Y. Kawamura, M.N., H.K., H.A., Y.W., T.F., and Y. Kanki analyzed the data; and Y.H., N. Nagai, T.K., and Y. Kanki wrote the manuscript.

## Competing interests

The authors declare no competing interests.
