## [Peer Review File · Communications Biology]

Reviewers' comments:

Reviewer #1 (Remarks to the Author):

In this study, Yoshiki et al. constructed Kdm7a KO mice by CRIPR/Cas9-mediated gene targeting. The authors clearly observed homeotic transformation of the mouse axial skeleton and altered Hox expression, which are the major and most intriguing findings of this study. The phenotype also supports the biological importance of histone demethylation. The authors also tried to understand the mechanisms underlying this phenotype, they observed the levels of H3K9me2 on posterior Hox genes are moderately increased in the Kdm7a KO embryos. However whether the H3K9me2 is the dominant factor that was altered in Kdm7a KO and led to Hox misregulation will need further validation during the revision.

Major concern,

Kdm7a is a dual demethylase possessing the ability to demethylate H3K9me2 and H3K27me2. Importantly, although understudied, H3K27me2 might be a more important histone mark regulating Hox genes. Either H3K27me2 level or H3K27me3 level indirectly affected by impaired K27me2 demethylation might be altered in Kdm7a KO. The authors should examine the changes of both H3K27me2 and H3K27me3 at Hox loci by ChIP-qPCR and ChIP-seq.

Minor concerns,

- 1) Western blot of Kdm7a should be performed to prove the loss of Kdm7a protein in the KO animals.
- 2) In Figure 3D and 3H, error bars should be calculated and added.
- 3) If ChIP'able antibody is available, Kdm7a ChIP-qPCR or seq should be done. These can demonstrate a direct mode of action of Kdm7a over Hox loci.

Reviewer #2 (Remarks to the Author):

In this manuscript, Higashijima et al. investigate the role of a lysine demethylase, Kdm7a, during mouse embryonic development. They use CRISPR/Cas9 to generate mice harbouring mutations of the gene, characterizing their axial skeletal morphology, expression of Hox genes, and levels of H3K9me2 across Hox clusters.

3 main claims are made:

- i. The mutant mice exhibit an anterior homeotic transformation of the axial skeleton.
- ii. Posterior Hox genes show lower levels of expression in mutant embryos compared to wild-type.
- iii. Levels of H3K9me2 are increased in the Hox clusters of mutant embryos compared to wild-type.

This is an interesting study, since Hox gene expression during development is correlated with dynamic changes in the distribution of histone modifications across the Hox clusters. Perturbation of histone demethylases in some cases leads to Hox-like axial phenotypes. The claims made in this manuscript are novel and suggest that Kdm7a is such a Hox regulator. This would represent an important finding for our understanding of how histone marks are altered along the Hox clusters during development and how this relates to the regulation of Hox gene expression and axial patterning. However, I have some concerns about the manuscript that should be addressed to strengthen the claims and to clarify the presentation.

Major points:

- i. Characterisation or prediction of the mutant Kdm7a protein. It is good that two different genetic backgrounds were used to generate mutants. However, the mutant Kdm7a protein is not characterized. Since the mutations are within the JmjC domain, it seems to be assumed that this produces a non-functional protein product. However, it is possible that the mutations generate a truncated form/s of Kdm7a. If so, could this be hypomorphic? These possibilities should be

discussed, and the mutant protein product should be predicted based on the mRNA sequence.

- ii. Presentation of the mutant phenotypes. In Fig1E-F, it will be difficult for readers who are unfamiliar with mouse axial patterning to fully appreciate the morphological differences between wild-type and mutant phenotypes. It would help if these images were made larger and the thoracic vertebrae also labelled.
- iii. Spatial analysis of Hox expression. While changes in expression levels of Hox genes are presented in Fig2G-H, it could also be possible that spatial expression domains differ in the mutants and that these are contributing to their altered axial morphology. Thus, analysis of spatial expression for some of the posterior Hox genes (such as those that show the greatest changes in expression levels) should be performed to address this possibility.
- iv. Statistical analysis of changes in H3K9me2 on Hox loci. For the findings presented in Fig3, it is difficult to infer whether the differences between wild-type and mutant are meaningful without any statistical analyses. For example, H3K9me2 levels appear to be low in the head and trunk, so are the differences between WT and KO at the Hox loci (Fig3H) statistically significant? If not, the claim that H3K9me2 levels are altered in the mutant is not convincing. Error bars and p-values should be provided for Fig3D,H to clarify this.
- v. Presentation of a model to summarise the findings. This manuscript would benefit from the inclusion of a final figure with a model depicting the inferred influence of Kdm7a on the H3K9me2 levels across the Hox clusters during development and how this relates to Hox expression and axial patterning.

Minor points:

- i. Line 110. Please provide a reference for the developmental brain defects in zebrafish.
- ii. Line 381. Fig1E. Co1 in legend is labelled Ca1 in Figure.
- iii. Figure 3D. Labelled as 'somite' and 'brain' but do they mean 'trunk' and 'head'?

Reviewer #3 (Remarks to the Author):

Higashijima and colleagues present the characterisation of skeletal defects, transcriptomic changes and altered histone methylation in Lysine demethylase 7a mutant mice. Skeletal characterisation and altered Hox signatures identified are novel advances for the field. I have questions regarding methylation that need to be addressed.

1. Mutant mouse generation and skeletal characterisation

Mutant generation is straightforward and phenotypes confirmed using two separate deletion lines, each on unique genetic backgrounds. Overall, the skeletal characterisation and experimental numbers appear appropriate. However the images, at least in the document I have, are fuzzy and unclear. Yes I can see T14, but any other detail of vertebral morphology is unclear. Can the authors please provide higher quality images. I am unsure what is meant by "the presence of cervical vertebral phenotypes in the thoracic vertebrae" or what the arrow in Fig 2E is pointing to. If reimaging does not help in presenting this cervical/thoracic data, I suggest the authors dissect and photograph individual vertebral elements to clearly demonstrate (for example see Wellik, Curr Topics in Dev Biol, 2009).

2. Altered molecular signature and Hox code

The authors go perform a global transcriptomic analysis following Kdm7a deletion, using a highly heterogeneous source of tissue. A delay in posterior Hox activation is observed which, consistent with extensive literature, has the potential to drive phenotypic changes.

3. Altered histone modifications following Kdm7a deletion.

A direct link between Kdm7a deletion-H3K9me2 maintenance-altered Hox expression is suggested, though the authors are careful not to state a causal link. However I have questions over the H3K9me2 data in Figure 3 B/C. Can the authors please clarify how peaks have been linked to specific Hox genes. Further to this, the graphical overlay of H3K9me2 difference between trunk

and head in Figure 3C does not appear to show the fold upregulation that is indicated in Figure 3B (ie a consistent 2-4 fold upregulation across the Hoxa cluster?). Can the authors please clarify this dataset.

Can the authors also please assess other histone modifications that Kdm7a has been suggested to impact, to provide a comprehensive understanding of how deletion of this gene affects morphology.

Additional comments:

Characterisation of the in situ expression of Kdm7a during mouse embryogenesis, particularly at the stages relevant to skeletal patterning, would be useful.

Referencing:

I believe some additional references would be appropriate and, in general, references should be placed at end of the first sentence where literature is mentioned.

- In the first paragraph, original refs for colinearity, RA/Wnt/FGF should be inserted.
- Pg 4. Kdm7a is predominantly expressed in mouse brain tissues – please reference.
- Inhibition of a Kdm7a ortholog in zebrafish leads to developmental brain defects (ref here, not following sentence).
- In addition, Kdm7a promotes neural differentiation of mouse ES cells by transcriptional activation of Fgf4, a signal molecule implicated in neural differentiation (ref here, not following sentence).

Minor text comments:

Pg 4, line 131: "Because there are no suitable protospacer-adjacent motif (PAM) sequences in exon5 of the region encoding the JmjC domain, we designed single guide RNAs (sgRNAs) located in exon6 of the region encoding the JmjC domain (Figure 1A)". This sentence is a bit confusing, Figure 1 indicates exon 5 does not encode JmjC domain (as written above).

Figures 1A and D - Details of the guide RNA sequence ultimately used for KO generation, and deletion characterisation is necessary. Fig 1B is unnecessary. This info can go to Supp Info if the authors feel strongly to include, as can Fig 1C. More important is a statement of what impact these deletions have on the resultant protein (ie. all are frameshift mutations, leading to premature truncation in all cases?).

Pg6: I'm not sure what is meant by "and this was evident in an E9.5 comparison with E10.5 (Figure 2H)" I don't believe the authors are comparing E9.5 with E10.5 here?

Pg 7: "In the last decades, histone-modifying enzymes have become recognized as the key players of the development, differentiation, and various diseases." Check grammar.

Reviewer #4 (Remarks to the Author):

Review COMMSBIO-19-1195

The manuscript of Higashijima et al., entitled « Lysine demethylase 7a regulates the anterior-posterior development in mouse by modulating the transcription of the Hox gene cluster » aimed at deciphering a new role of the histone demethylase Kdm7a during vertebrate development, possibly via its catalytic activity on H3K9me2. In particular, they explored the question of the regulation of the transcriptional expression of the master regulators of the A/P axis identity during development, the well conserved family of Hox TFs. They showed that Kdm7a knock out is associated with anterior homeotic transformation of the axial skeleton, a decrease of posterior Hox

genes expression and an increased of H3K9me2 enrichment. All in all, it suggests that Kdm7a regulates Hox genes expression via its demethylase activity on H3K9me2, thereby controlling embryo patterning.

The novelty concerning the role of Kdm7a in development and patterning and the importance for further researches on the regulation of the expression of Hox gene at the chromatin level is of particular relevance. Furthermore, the question is elegantly tackled at in vivo level using CRISPR/Cas9, as well as the analysis of different mutations in two distinct genetic backgrounds. It is strengthened by transcriptomic and genomic approaches performed at the in vivo level in mouse embryos.

However, the paper does not always provide strong evidence and lacks more in-depth analysis to re-enforce the potential link between Hox expression and the H3K9me2 demethylation role of Kdm7a. In particular, genome-wide profiles by transcriptomic & genomic approaches should be studied more in depth as considerable information is left out (expression of other demethylases, methyltransferases, histone modifiers in general as well as regulators of Hox gene expression such as PcG, TrxG proteins groups). In line, the technical information & validation of the replicates are absent and undoubtedly required to evaluate properly the significance of the data. Moreover, there are neither evidences nor hypothesis concerning the role of Kdm7a on H3K27me2, which might be required for Hox gene regulation (or not). In line, the reason why the author only focused on H3K9me2 is not explained. All in all, it weakens the conclusions drawn by the manuscript and the take-home message.

Thus, the paper requires major revisions to be considered for publication, in particular, more in depth bioinformatics analysis of the data generated and few additional experiments to re-enforce the general conclusion of the manuscript.

Major comments:

-Figure 1E

Different deletions were generated in different genetic backgrounds. It might drive different level of (residual or truncated) expression of Kdm7a that might be the reason for the distinct penetrance of the phenotype in ICRvsC57BL/6 and not only the background (as suggested Page5, lines 172-174). Thus, the residual expression of Kdm7a should be controlled and shown, by Western-blot for example.

-In line, the distinct penetrance of the phenotype in heterozygous ICR or C57BL/6 may be discussed.

-Genomic and transcriptomic analysis:

More information in the main text concerning replicates (source, number) and validation (comparison by Pearson correlation for example)

-Detailed analysis of RNAseq should be shown:

-Upon Kdm7a KO, expression of all histone modifiers (methyltransferases such as G9a and demethylases...) with a particular focus on K9me (and K27me) modifiers, regulators of Hox clusters, Polycomb & trithorax protein family expression

-Very few numbers of genes are deregulated. This could be discussed. Might it be changed if stringency parameters would be adjusted? Is it relevant? Could the authors show both analyses (stringent and milder)? In line, few genes are deregulated by RNAseq while analysis by qPCR revealed a difference in expression in control versus mutant.

-High difference can be noticed between E9.5 and E10.5, for which only 2 genes are misexpressed. This could be discussed more in depth in particular as E10.5 is chosen for the following ChIPseq study.

-Why did the author not perform the comparison between head and trunk too to confirm the

differential expression?

-Detailed analysis of ChIP-seq:

-Idem as RNAseq for replicates description & validation

-The comparison between head and trunk is a bit confusing. The biological question might be clarified to help the reader. It sounds like a characterization of the Hox inactive locus in the head versus active in the trunk.

-Page 7, line 221-22: the sentence is confusing. It is not an increase of H3K9me2 but a higher enrichment of H3K9me2 in the head compared to the trunk, that is in line with the inactive state of the Hox genes in the head (and not in the trunk). Reformulating the sentence would be helpful for the clarity of the message driven by the data.

-In line Fig 3D: somite/brain should be also related to head/trunk for helping the reader.

-Moreover, interpreted conclusions could be added in the text for guiding the reader

-The state of H3K9me2 is enriched in the head if the genes are already downregulated/not expressed. This might be relevant to link this point with the choice of not doing the RNAseq on the head.

-Fig 3F: a large number of anterior Hox genes present an increase of H3K9me2 enrichment. This might be discussed.

-Figure 3D and 3H:

The error bars for the ChIP-qPCR are required.

It seems that H3K4me3 is more enriched for the mutant compared to control for Hoxa13 (which is in opposition with H3K9me2 repressive mark accumulation). It might be useful to change the normalization to Histone3 in this way.

-H3K27me2 regulation:

Finally, concerning the role of Kdm7a on K27me2, some additional experiments, at least by ChIP-qPCR on selected Hox targets (Hox3, Hox13...) and controls (at least two, known positive & negative) should be provided (KOsControl).

Minor comments

-Page 4, line138 & figure 1:

Quantification of signal generated upon expression of sgRNA868/867 in HeLa cells control experiments could be provided

-Figure1D, RFLP: legend of mice genetic background is missing

-Figure 2A: expression pattern of the Hox might be drawn on the cartoon to correlate with the embryos cutting and help the reader in the lecture

-Figure 2D left panel second graphical view: Orthography: "comoplex"

-Figure 3F: the case of Hox12c might be discussed

-Figure 3F: It would help the reader if anterior and posterior Hox are indicated.

-Figure 3: H3K9me2 and K4me3 should be normalized to H3 or at least, H3 ChIP-qPCR could be shown

-Page9, line297-301: it seems not in line with the result showing that Kdm7a KO is associated with a decrease of Hox gene expression. The result suggests that Kdm7a is an activator. The hypothesis developed links the repressive mark K27me3 and nuclear active environment that seems confusing. Reference should be added as well as the idea behind clarified.

-Page9, line 302-315: this part is based on assumption and should be nuanced. The link between the different studies needs to be re-enforced. Alternatively, this part may be removed as it sounds lacking strength.

Author's responses to the reviewer's comments

Reviewer #1 (Remarks to the Author):

In this study, Yoshiki et al. constructed Kdm7a KO mice by CRIPR/Cas9-mediated gene targeting. The authors clearly observed homeotic transformation of the mouse axial skeleton and altered Hox expression, which are the major and most intriguing findings of this study. The phenotype also supports the biological importance of histone demethylation. The authors also tried to understand the mechanisms underlying this phenotype, they observed the levels of H3K9me2 on posterior Hox genes are moderately increased in the Kdm7a KO embryos. However whether the H3K9me2 is the dominant factor that was altered in Kdm7a KO and led to Hox misregulation will need further validation during the revision. Major concern, Kdm7a is a dual demethylase possessing the ability to demethylate H3K9me2 and H3K27me2. Importantly, although understudied, H3K27me2 might be a more important histone mark regulating Hox genes. Either H3K27me2 level or H3K27me3 level indirectly affected by impaired K27me2 demethylation might be altered in Kdm7a KO. The authors should examine the changes of both H3K27me2 and H3K27me3 at Hox loci by ChIP-qPCR and ChIP-seq.

We are grateful for the reviewer's comments and suggestions which have significantly help to improve our manuscript. In response to the points and suggestions raised by the reviewer, we have revised the manuscript. Our point-by-point responses to the reviewer's comments are as follows:

In accordance with the reviewer's suggestion, we have performed ChIP-seq for H3K27me2 and H3K27me3. As a result, consistent with a previous report, we observed, in developmental trunk (where Hox genes are actively transcribed), the entire deposition of an active histone mark H3K4me3 at a representative *Hoxa* cluster which was paralleled by relatively low enrichment of repressive histone mark H3K27me3. Conversely, H3K27me3 covered the entire *Hoxa* gene cluster in the developmental head (where Hox genes are inactive) which was associated with extremely low enrichment of H3K4me3 (Please see revised Supplementary Fig. 5b and c). Importantly, we observed a higher enrichment of H3K9me2 in the head compared to that in the trunk at the representative *Hoxa* cluster, while these for H3K27me2 were not altered (Please see Supplementary Fig. 5d and e). Furthermore, we have additionally performed ChIP-qPCR for H3K27me2 and H3K27me3. ChIP-qPCR confirmed an increase in H3K9me2 and H3K27me3 levels and a decrease in H3K4me3 levels at *Hoxa3* and *Hoxa13* loci. Nevertheless, we detected no differences in H3K27me2 between the regions of head and trunk (Please see revised Supplemental Fig. 5e), suggesting that H3K9me2 (also H3K4me3 and H3K27me3), but not

H3K27me2, could be associated with transcriptional state of *Hox* gene locus during mouse embryonic development. We next examined whether ablation of *Kdm7a* affected the epigenetic landscape at the *Hox* genes in the developmental trunk regions. ChIP-seq analysis demonstrated relatively high occupancy of H3K9me2 at the representative *Hoxa* locus in the *Kdm7a*^{-/-} embryonic trunk compared to wild-type, but no differences in the levels of H3K27me2 (Please see revised Fig. 3b and c). Despite the fact that opposed labeling of H3K4me3 and H3K27me3 is involved in the regulation of *Hox* genes during development, there were no obvious differences in the levels of H3K4me3 and H3K27me3 between the wild-type and *Kdm7a*^{-/-} embryonic trunk (Please see Fig. 3b and c). Consistently, ChIP-qPCR showed an increase in H3K9me2 levels at the *Hoxa3* and *Hoxa13* loci, but no changes at the *Actb* locus (Please see Fig. 3e). We detected no differences of H3K4me3, H3K27me2, and H3K27me3 levels between the wild-type and the *Kdm7a*^{-/-} trunk at the *Actb*, *Hoxa3*, and *Hoxa13* loci during the ChIP-qPCR analysis (Please see Fig. 3e). Taken together, we believe the possibility that *Kdm7a*-mediated regulation of the repressive histone mark H3K9me2, but not H3K27me2, might be involved in transcriptional activation of the *Hox* genes. We have included these results in the revised manuscript, (Page 7, Line 225-266) In addition, we have also added the discussion regarding the role of H3K27me2 in *Hox* gene regulation, (Page 10, Line 322-338).

Minor concerns,

1) Western blot of *Kdm7a* should be performed to prove the loss of *Kdm7a* protein in the KO animals.

Thank you for your suggestion. Accordingly, we have performed western blot analysis for detecting *Kdm7a* protein in the both wild-type and *Kdm7a*^{-/-} mouse. The sample was extracted from mouse brain tissue in which *Kdm7a* protein was reportedly predominantly expressed¹. We have tried various antibodies against *Kdm7a* from LsBio (targeting 290AA-316AA), abcam (targeting 750AA-850AA), Novus (targeting C-terminal), Millipore (polyclonal antibody; without specific information), and self-produced antibody (a kind gift from Prof. Tsukada, Kyusyu University). However, unfortunately, we could not obtain reproducible and reliable results from western blot analysis although we could detect overexpressed *Kdm7a* protein in cultured cells (data not shown). Thus, we believe that detecting *Kdm7a* protein in mice tissue is particularly challenging at this moment. Alternatively, we have added detailed information regarding the protein resulted in the *Kdm7a* mutant mice (please see revised Fig. 1b). All mutant mice carried frameshift mutations, resulting in truncated *Kdm7a* proteins that lack core catalytic amino acid for its demethylase activity (His²⁸⁴ at Fe(II)-binding site)¹. We have included this detailed information in the result section of the revised manuscript, (Page 5, Line

151-153).

2) In Figure 3D and 3H, error bars should be calculated and added.

Thank you for your suggestion. We have calculated and added error bars. Please see revised Fig. 3e and Supplementary Fig. 5e.

3) If ChIP'able antibody is available, Kdm7a ChIP-qPCR or seq should be done. These can demonstrate a direct mode of action of Kdm7a over Hox loci.

Thank you for your suggestion. As mentioned above, unfortunately, we could not obtain good antibody for even western blot analysis or ChIP experiments.

Reviewer #2 (Remarks to the Author):

In this manuscript, Higashijima et al. investigate the role of a lysine demethylase, Kdm7a, during mouse embryonic development. They use CRISPR/Cas9 to generate mice harbouring mutations of the gene, characterizing their axial skeletal morphology, expression of Hox genes, and levels of H3K9me2 across Hox clusters. 3 main claims are made:

- i. The mutant mice exhibit an anterior homeotic transformation of the axial skeleton.
- ii. Posterior Hox genes show lower levels of expression in mutant embryos compared to wild-type.
- iii. Levels of H3K9me2 are increased in the Hox clusters of mutant embryos compared to wild-type.

This is an interesting study, since Hox gene expression during development is correlated with dynamic changes in the distribution of histone modifications across the Hox clusters.

Perturbation of histone demethylases in some cases leads to Hox-like axial phenotypes. The claims made in this manuscript are novel and suggest that Kdm7a is such a Hox regulator. This would represent an important finding for our understanding of how histone marks are altered along the Hox clusters during development and how this relates to the regulation of Hox gene expression and axial patterning. However, I have some concerns about the manuscript that should be addressed to strengthen the claims and to clarify the presentation.

We are grateful for the reviewer's comments and suggestions which have significantly help to improve our manuscript. In response to the points and suggestions raised by the reviewer, we have revised our manuscript. Our point-by-point responses to the reviewer's comments are as follows:

Major points:

i. Characterisation or prediction of the mutant Kdm7a protein. It is good that two different genetic backgrounds were used to generate mutants. However, the mutant Kdm7a protein is not characterized. Since the mutations are within the JmjC domain, it seems to be assumed that this produces a non-functional protein product. However, it is possible that the mutations generate a truncated form/s of Kdm7a. If so, could this be hypomorphic? These possibilities should be discussed, and the mutant protein product should be predicted based on the mRNA sequence.

Thank you for your comment and question. Accordingly, we have added detailed information regarding the protein resulted in the Kdm7a mutant mice (please see revised Fig. 1b). All mutant mice were carrying frameshift mutations, which resulted in truncated Kdm7a proteins that lacked core catalytic amino acid for its demethylase activity (His²⁸⁴ at Fe(II)-binding site)¹. Thus, we believe that truncated Kdm7a proteins are hypo-morphic. We have included this detailed information in the result section. In addition, we have performed western blot analysis for detecting Kdm7a protein in the both wild-type and *Kdm7a*^{-/-} mouse. The sample was extracted from mouse brain tissue in which Kdm7a protein was reportedly predominantly expressed. We have tried various antibodies against Kdm7a from LsBio (targeting 290AA-316AA), abcam (targeting 750AA-850AA), Novus (targeting C-terminal), Millipore (polyclonal antibody; without specific information), and self-produced antibody (a kind gift from Prof. Tsukada, Kyusyu University), however, unfortunately, we couldn't obtain reproducible and reliable results from the western blot analysis, although we could detect overexpressed Kdm7a protein in cultured cells. Thus, we think that detecting Kdm7a protein in mouse tissue is challenging at this moment.

ii. Presentation of the mutant phenotypes. In Fig1E-F, it will be difficult for readers who are unfamiliar with mouse axial patterning to fully appreciate the morphological differences between wild-type and mutant phenotypes. It would help if these images were made larger and the thoracic vertebrae also labelled.

Thank you for your suggestion. We have prepared high resolution and enlarged images and labelled the thoracic vertebrae. Please see the revised Fig. 1c-e. To help the readers better understanding and focusing on anterior-posterior shift, we have removed the description of “the presence of cervical vertebral phenotypes in the thoracic vertebrae”, and alternatively, we have also prepared enlarged images of thoracolumbar.

iii. Spatial analysis of Hox expression. While changes in expression levels of Hox genes are presented in Fig2G-H, it could also be possible that spatial expression domains differ in the mutants and that these are contributing to their altered axial morphology. Thus, analysis of spatial expression for some of the posterior Hox genes (such as those that show the greatest changes in expression levels) should be performed to address this possibility.

Thank you for your suggestion. Accordingly, we have performed whole mount *in situ* hybridization of representative posterior Hox genes (i.e. *Hoxd9* and *Hoxd10*); as the revised Fig. 2h shows, although the transcript levels of *Hoxd9* and *Hoxd10* were markedly decreased in the *Kdm7a*^{-/-} embryos compared with the wild-type, their spatial distribution was not altered. We have included this in the result section of the revised manuscript, (Page 6, Line 211-216).

iv. Statistical analysis of changes in H3K9me2 on Hox loci. For the findings presented in Fig3, it is difficult to infer whether the differences between wild-type and mutant are meaningful without any statistical analyses.

For example, H3K9me2 levels appear to be low in the head and trunk, so are the differences between WT and KO at the Hox loci (Fig3H) statistically significant? If not, the claim that H3K9me2 levels are altered in the mutant is not convincing. Error bars and p-values should be provided for Fig3D,H to clarify this.

Thank you for your suggestion. We have calculated and added error bars. As shown in revised Fig. 3e and supplementary Fig. 5e, H3K9me2 level at *Hoxa13* loci was significantly increased in the *Kdm7a*^{-/-} mice compared to wild-type ($P < 0.038$). Also, H3K9me2 level at *Hoxa3* loci was significantly increased in the developmental head compared to the trunk ($P < 0.01$). In addition, we performed ChIP-seq analysis with two biological replicates (with high correlation, please see the revised Supplementary Fig. 4) and provided average heatmap of ChIP-seq signals for H3K9me2 (please see revised Fig. 3d and Supplementary Fig. 5d). These heatmaps also demonstrated increased H3K9me2 levels in the *Kdm7a*^{-/-} embryos and the developmental head. Together, we believe that H3K9me2 could be associated with the regulation of Hox gene expression.

v. Presentation of a model to summarise the findings. This manuscript would benefit from the inclusion of a final figure with a model depicting the inferred influence of Kdm7a on the H3K9me2 levels across the Hox clusters during development and how this relates to Hox expression and axial patterning.

Thank you for your suggestion. We have added a graphical model to summarize our findings. Please see the revised Fig. 4.

Minor points:

i. Line 110. Please provide a reference for the developmental brain defects in zebrafish.

Thank you for your comment. We have provided a reference regarding this reviewer's comment¹.

ii. Line 381. Fig1E. Co1 in legend is labelled Ca1 in Figure.

Thank you for your comment. We have rephrased "Ca1" to "Co1" in the revised Fig. 1.

iii. Figure 3D. Labelled as 'somite' and 'brain' but do they mean 'trunk' and 'head'?

Thank you for your question. We have rephrased "somite" and "brain" to "trunk" and "head", respectively in revised supplementary Fig. 5e.

Reviewer #3 (Remarks to the Author):

Higashijima and colleagues present the characterisation of skeletal defects, transcriptomic changes and altered histone methylation in Lysine demethylase 7a mutant mice. Skeletal characterisation and altered Hox signatures identified are novel advances for the field. I have questions regarding methylation that need to be addressed.

We are grateful for the reviewer's comments and suggestions which have significantly help to improve our manuscript. In response to the points and suggestions raised by the reviewer, we have revised our manuscript. Our point-by-point responses to the reviewer's comments are as follows:

1. Mutant mouse generation and skeletal characterisation

Mutant generation is straightforward and phenotypes confirmed using two separate deletion lines, each on unique genetic backgrounds. Overall, the skeletal characterisation and experimental numbers appear appropriate. However the images, at least in the document I have, are fuzzy and unclear. Yes I can see T14, but any other detail of vertebral morphology is unclear. Can the authors please provide higher quality images. I am unsure what is meant by

“the presence of cervical vertebral phenotypes in the thoracic vertebrae” or what the arrow in Fig 2E is pointing to. If reimaging does not help in presenting this cervical/thoracic data, I suggest the authors dissect and photograph individual vertebral elements to clearly demonstrate (for example see Wellik, Curr Topics in Dev Biol, 2009).

Thank you for your suggestion and assessment. Accordingly, we have prepared high resolution images and newly labelled the thoracic vertebrae. Please see the revised Fig. 1c-e. To help the readers better understanding and focusing on anterior-posterior shift, we have removed the description of “the presence of cervical vertebral phenotypes in the thoracic vertebrae”, and alternatively, we have also prepared enlarged images of thoracolumbar.

2. Altered molecular signature and Hox code

The authors go perform a global transcriptomic analysis following *Kdm7a* deletion, using a highly heterogeneous source of tissue. A delay in posterior Hox activation is observed which, consistent with extensive literature, has the potential to drive phenotypic changes.

Thank you for your comment. We performed a global transcriptomic analysis using a heterogeneous source of tissue (please see “Methods section: Dissection of the anterior and posterior parts of the embryo”) and found that posterior Hox genes were preferentially down-regulated in the *Kdm7a*^{-/-} compared to wild-type

3. Altered histone modifications following *Kdm7a* deletion.

A direct link between *Kdm7a* deletion-H3K9me2 maintenance-altered Hox expression is suggested, though the authors are careful not to state a causal link. However I have questions over the H3K9me2 data in Figure 3 B/C. Can the authors please clarify how peaks have been linked to specific Hox genes. Further to this, the graphical overlay of H3K9me2 difference between trunk and head in Figure 3C does not appear to show the fold upregulation that is indicated in Figure 3B (ie a consistent 2-4 fold upregulation across the *Hoxa* cluster?). Can the authors please clarify this dataset. Can the authors also please assess other histone modifications that *Kdm7a* has been suggested to impact, to provide a comprehensive understanding of how deletion of this gene affects morphology.

Thank you for your comment and suggestion. We have performed ChIP-seq analysis with two biological replicates and confirmed that our analysis was highly correlated (Please see revised Supplementary Fig. 4). Then, since it is challenging to annotate broad H3K9me2 peaks to specific Hox genes, which form dense clusters, we have recalculated the average log2 fold

change of ChIP-seq signals based on READ COUNTS of H3K9me2 for each Hox gene, and visualized them as heatmaps (Please see revised Fig. 3d (KO/WT) and Supplementary Fig 5d (Head/trunk)). Briefly, we calculated log₂ fold change of ChIP-seq signals using the following formula:

$$\log_2 \left\{ \left(\frac{\text{CPM}^{\text{IP1}}}{\text{CPM}^{\text{input1}}} \right) / \left(\frac{\text{CPM}^{\text{IP2}}}{\text{CPM}^{\text{input2}}} \right) \right\}$$

where CPM^{IP} and $\text{CPM}^{\text{input}}$ represent the read counts per million mapped reads (CPM) from IP and input libraries, respectively; we have provided detailed information in the Methods section.

Important point to note here is that the values used for heatmaps were normalized with the read count of each input sample. The discrepancy between presentation of the IGV browser and the heatmaps may be caused by a technical limitation for producing ChIP-seq signal tracks. We normally do not use input normalization to show our ChIP-seq dataset on the IGV browser as is in previous studies. In cross-sample comparisons between ChIP-seq signal tracks, signals need to be normalized with the number of mapped reads. This normalization, however, narrows differences between samples in cases where the signal is globally increased or decreased over the genome compared to another sample. In our study, *Kdm7a* knock-out is expected to result in genome-wide enrichment of H3K9me2 as well as locus-specific changes, which should be masked by normalization using total mapped read counts. Hence, presentation of the IGV browser possibly underestimates the differences between WT and KO. To confer this, in generating heatmap, we normalized ChIP-seq signals using background signals (i.e. input library) as standards. That is why, the presence or absence of normalization with input might explain the discrepancy between presentation of the IGV browser and the heatmaps, although we could observe only a mild change of the appearance of heatmaps when calculated without input-normalization (data not shown). Nevertheless, we could observe the increase of H3K9me2 in the *Kdm7a*^{-/-} embryos compared to the wild-type using two different methods (i.e. ChIP-seq and ChIP-qPCR, please see revised Fig. 3), and therefore we believe the possibility that *Kdm7a*-mediated regulation of the repressive histone mark H3K9me2 might be involved in the transcriptional activation of the *Hox* genes.

Accordingly, we have performed ChIP-seq for H3K27me2 and H3K27me3. As a result, consistent with a previous report², we observed, in the developmental trunk (where Hox genes are actively transcribed), the entire deposition of an active histone mark H3K4me3 at a representative *Hoxa* cluster which was paralleled by relatively low enrichment of repressive histone mark H3K27me3. Conversely, H3K27me3 covered the entire *Hoxa* gene cluster in the developmental head (where Hox genes are inactive) which was associated with extremely low enrichment of H3K4me3 (Please see revised Supplementary Fig. 5b and c). Importantly, we observed a higher enrichment of H3K9me2 in the head compared to the trunk at the representative *Hoxa* cluster, while these for H3K27me2 were not altered. Furthermore, we have

additionally performed ChIP-qPCR for H3K27me2 and H3K27me3. ChIP-qPCR confirmed an increase in H3K9me2 and H3K27me3 levels and a decrease in H3K4me3 levels at *Hoxa3* and *Hoxa13* loci. Nevertheless, we detected no differences in H3K27me2 between the regions of head and trunk (Please see revised Supplemental Fig. 5e), suggesting that H3K9me2 (as well as H3K4me3 and H3K27me3) but not H3K27me2 could be associated with the transcriptional state of *Hox* gene locus during mouse embryonic development. We next examined whether ablation of *Kdm7a* affected the epigenetic landscape at the *Hox* genes in the developmental trunk regions. ChIP-seq analysis demonstrated relatively high occupancy of H3K9me2 at the representative *Hoxa* locus in the *Kdm7a*^{-/-} embryonic trunk compared with the wild-type, but no differences were detected in the levels of H3K27me2 (Please see revised Fig. 3b and c). Despite the fact that opposed labeling of H3K4me3 and H3K27me3 is involved in the regulation of *Hox* genes during development, there were no obvious differences in the levels of H3K4me3 and H3K27me3 between the wild-type and *Kdm7a*^{-/-} embryonic trunk (Please see Fig. 3b and c). Consistently, ChIP-qPCR showed an increase in H3K9me2 levels at *Hoxa3* and *Hoxa13* loci, but no changes at the *Actb* locus (Please see Fig. 3e). We detected no differences in H3K4me3, H3K27me2, and H3K27me3 levels between the wild-type and the *Kdm7a*^{-/-} trunk at the *Actb*, *Hoxa3*, and *Hoxa13* loci during the ChIP-qPCR analysis (Please see Fig. 3e). Taken together, we believe the possibility that *Kdm7a*-mediated regulation of the repressive histone mark H3K9me2, but not H3K27me2, might be involved in the transcriptional activation of the *Hox* genes. We have included these results in the revised manuscript, (Page 7, Line 225-268).

Additional comments:

Characterisation of the *in situ* expression of *Kdm7a* during mouse embryogenesis, particularly at the stages relevant to skeletal patterning, would be useful.

According to the reviewer's suggestion, we have performed whole mount *in situ* hybridization of *Kdm7a* mRNA in the embryos. In wild-type embryos at E8.5, the expression of *Kdm7a* was observed in the primitive streak and presomitic mesoderm, where *Hox9-10* genes were started to be activated during development³ (Please see supplementary Fig. 3). We have included this information in the result section of the revised manuscript. This could be consistent with a previous report showing *Kdm7a* began to express in developmental head and tailbud of zebrafish at 24 post-fertilization, which is corresponding to E8.5 of mice¹. We have included this information in the result section of the revised manuscript, (Page 7, Line 216-221).

Referencing:

I believe some additional references would be appropriate and, in general, references should be

placed at end of the first sentence where literature is mentioned.

- In the first paragraph, original refs for colinearity, RA/Wnt/FGF should be inserted.
- Pg 4. Kdm7a is predominantly expressed in mouse brain tissues – please reference.
- Inhibition of a Kdm7a ortholog in zebrafish leads to developmental brain defects (ref here, not following sentence).
- In addition, Kdm7a promotes neural differentiation of mouse ES cells by transcriptional activation of Fgf4, a signal molecule implicated in neural differentiation (ref here, not following sentence).

Thank you for your comment. We have provided the relevant references accordingly^{1,4,5}.

Minor text comments:

Pg 4, line 131: “Because there are no suitable protospacer-adjacent motif (PAM) sequences in exon5 of the region encoding the JmjC domain, we designed single guide RNAs (sgRNAs) located in exon6 of the region encoding the JmjC domain (Figure 1A)”. This sentence is a bit confusing, Figure 1 indicates exon 5 does not encode JmjC domain (as written above).

Thank you for your comment. As we described in the original manuscript, JmjC domain starts from the end of exon5 and stops at the beginning of exon9. We have prepared a high resolution figure with minor modification to better illustrate the start of the JmjC domain.

Figures 1A and D - Details of the guide RNA sequence ultimately used for KO generation, and deletion characterisation is necessary. Fig 1B is unnecessary. This info can go to Supp Info if the authors feel strongly to include, as can Fig 1C. More important is a statement of what impact these deletions have on the resultant protein (ie. all are frameshift mutations, leading to premature truncation in all cases?).

Thank you for your suggestion. We have moved the original Fig. 1b-d to the supplementary Fig. 1a-c. In addition, we have provided detailed information regarding the resulting protein in Kdm7a mutant mice (please see revised Fig. 1b). All mutant mice carried frameshift mutations, which resulted in truncated Kdm7a proteins that lacked core catalytic amino acid for its demethylase activity (His²⁸⁴ at Fe(II)-binding site)¹. We have included in the result section of the revised manuscript, (Page 5, Line 151-153).

Pg6: I'm not sure what is meant by “and this was evident in an E9.5 comparison with E10.5 (Figure 2H)” I don't believe the authors are comparing E9.5 with E10.5 here?

Thank you for your suggestion. Accordingly, we have removed the sentence “and this was evident in an E9.5 comparison with E10.5”.

Pg 7: “In the last decades, histone-modifying enzymes have become recognized as the key players of the development, differentiation, and various diseases.” Check grammar.

Thank you for your comment. We have rephrased the sentence as “Histone-modifying enzymes have been recognized as key players during early development and differentiation, as well as various diseases.” Please see Page 8, Line 271-272. In addition, the manuscript has been edited for English language and we have provided a certificate of proofreading.

Reviewer #4 (Remarks to the Author):

Review COMMSBIO-19-1195

The manuscript of Higashijima et al., entitled « Lysine demethylase 7a regulates the anterior-posterior development in mouse by modulating the transcription of the Hox gene cluster » aimed at deciphering a new role of the histone demethylase Kdm7a during vertebrate development, possibly via its catalytic activity on H3K9me2. In particular, they explored the question of the regulation of the transcriptional expression of the master regulators of the A/P axis identity during development, the well conserved family of Hox TFs. They showed that Kdm7a knock out is associated with anterior homeotic transformation of the axial skeleton, a decrease of posterior Hox genes expression and an increased of H3K9me2 enrichment. All in all, it suggests that Kdm7a regulates Hox genes expression via its demethylase activity on H3K9me2, thereby controlling embryo patterning. The novelty concerning the role of Kdm7a in development and patterning and the importance for further researches on the regulation of the expression of Hox gene at the chromatin level is of particular relevance. Furthermore, the question is elegantly tackled at in vivo level using CRISPR/Cas9, as well as the analysis of different mutations in two distinct genetic backgrounds. It is strengthened by transcriptomic and genomic approaches performed at the in vivo level in mouse embryos. However, the paper does not always provide strong evidence and lacks more in-depth analysis to re-enforce the potential link between Hox expression and the H3K9me2 demethylation role of Kdm7a. In particular, genome-wide profiles by transcriptomic & genomic approaches should be studied more in depth as considerable information is left out (expression of other demethylases, methyltransferases, histone modifiers in general as well as regulators of Hox gene expression such as PcG, TrxG proteins groups). In line, the technical information & validation of the replicates are absent and undoubtedly required to evaluate properly the significance of the data. Moreover, there are

neither evidences nor hypothesis concerning the role of Kdm7a on H3K27me2, which might be required for Hox gene regulation (or not). In line, the reason why the author only focused on H3K9me2 is not explained. All in all, it weakens the conclusions drawn by the manuscript and the take-home message. Thus, the paper requires major revisions to be considered for publication, in particular, more in depth bioinformatics analysis of the data generated and few additional experiments to re-enforce the general conclusion of the manuscript.

We are grateful for the reviewer's comments and suggestions which have significantly help to improve our manuscript. In response to the points and suggestions raised by the reviewer, we have revised our manuscript. Our point-by-point responses to the reviewer's comments are as follows:

Major comments:

Figure 1E

Different deletions were generated in different genetic backgrounds. It might drive different level of (residual or truncated) expression of Kdm7a that might be the reason for the distinct penetrance of the phenotype in ICRvsC57BL/6 and not only the background (as suggested Page5, lines 172-174). Thus, the residual expression of Kdm7a should be controlled and shown, by Western-blot for example.

In line, the distinct penetrance of the phenotype in heterozygous ICR or C57BL/6 may be discussed.

Thank you for your comment. We have performed western blot analysis for detecting Kdm7a protein in the both wild-type and *Kdm7a*^{-/-} mice. The sample was extracted from the brain tissue in which Kdm7a protein was reportedly predominantly expressed¹. We have tried various antibodies against Kdm7a from LsBio (targeting 290AA-316AA), abcam (targeting 750AA-850AA), Novus (targeting C-terminal), Millipore (polyclonal antibody; without specific information), and self-produced antibody (a kind gift from Prof. Tsukada, Kyusyu University); however, unfortunately, we couldn't obtain reproducible and reliable results from the western blot analysis, although we detected overexpressed Kdm7a protein in cultured cells (data not shown). Thus, we believe that detecting Kdm7a protein in mice tissue is challenging at this moment. Alternatively, we have added detailed information regarding the resulting protein in *Kdm7a* mutant mice (please see revised Fig. 1b). All mutant mice carried frameshift mutations, which caused truncated Kdm7a proteins that lacked core catalytic amino acid for its demethylase activity (His²⁸⁴ at Fe(II)-binding site)¹. As described in the main text, the phenotypes of mutant mice can differ between genetic backgrounds, especially for epigenetic

factors^{6,7}. Therefore, we believe that the phenotypic difference between ICR and B6 background could be caused by genetic background rather than resulting truncated proteins. We have included these in the result section of the revised manuscript.

Genomic and transcriptomic analysis: More information in the main text concerning replicates (source, number) and validation (comparison by Pearson correlation for example)

Detailed analysis of RNAseq should be shown:

Thank you for your comment. For genomic and transcriptomic analysis, the embryos dissected were from *Kdm7a*^{-/-} mice and their respective littermate control mice. The number of animals used in our studies has been described in the Figure Legends (i.e. n=3 for RNA-seq at E9.5 and E10.5, n=5 and 4 for qPCR at E9.5 and E10.5, respectively). These sentences have been included in the Methods and Figure Legends.

We have also added the correlation plot calculated by Pearson's r in the revised supplementary Fig. 2, and performed RNA-seq analysis with three biological replicates with a high correlation and confirmed the RNA-seq results using qPCR. Therefore, we believe that our RNA-seq result is reproducible and reliable.

Upon *Kdm7a* KO, expression of all histone modifiers (methyltransferases such as G9a and demethylases...) with a particular focus on K9me (and K27me) modifiers, regulators of Hox clusters, Polycomb & trithorax protein family expression

Thank you for your suggestion, we have checked the expression changes of all known histone methyltransferases and demethylases listed in the WERAM: a database of writers, erasers and readers of acetylation and methylation in eukaryotes (http://weram.biocuckoo.org/species.php?spe=Mus_musculus)⁸. As a result, only one gene, *Prdm12* (which contains an N-terminal SET domain), was down-regulated with >2-fold difference between the *Kdm7a*^{-/-} and the wild-type embryos (only at E9.5 but not E10.5). Nevertheless, down-regulation of *Prdm12* was not statistically significant (padj = 0.183, please see supplementary Table1, padj indicates false discovery rate). Of note, the expression of all other H3K9 and H3K27 histone methyltransferases and demethylases including *Jmjd1a*, *G9a*, *Ezh2*, and *Utx* were not altered, suggesting that the down-regulation of Hox genes observed in *Kdm7a*^{-/-} embryos might not be caused by a secondary effect of transcriptional changes in other histone methyltransferases or demethylases. We have included this information in the result section.

Very few numbers of genes are deregulated. This could be discussed. Might it be changed if stringency parameters would be adjusted? Is it relevant? Could the authors show both analyses (stringent and milder)? In line, few genes are deregulated by RNAseq while analysis by qPCR revealed a difference in expression in control versus mutant.

Thank you for your comment. We have added a newly prepared Supplementary Table1, which includes the expression value (TPM) of individual mice and the statistical tests results produced by DESeq2 for all genes. When we changed the statistical stringency from $\text{padj} < 0.05$ to $\text{padj} < 0.10$, the number of deregulated genes was changed from 73 to 96 in E9.5 embryos, and from 2 to 3 in E10.5 embryos. In our experience, RNA-seq analyses on samples, including various cell types such as tissues and embryos, shows that the repertoire of mRNA (i.e. the kinds of expressed genes) increases and therefore the read coverage, of each gene, relatively decreases. This lowers the sensitivity for statistical tests, while qPCR is robust against the effect of genes other than the target gene. This tendency is further pronounced in the E10.5 embryos, which is larger than that of the E9.5. To optimize a statistical test method for our RNA-seq data, we did not use log2 fold change ($\log_2\text{FoldChange}$ in Supplementary Table 1) as a cut-off value, which is a parameter widely used for identifying differentially expressed genes. Whether our RNA-seq analysis completely identifies all the differentially expressed transcripts in *Kdm7a* KO mice needs to be further investigated. Nevertheless, we observed the downregulation of Hox genes during the RNA-seq analysis with three biological replicates (with high correlation); qPCR analysis with multiple replicates (with statistical significance); and whole mount *in situ* hybridization (Please see Fig. 2f-h). Therefore, we believe that *Kdm7a* could regulate Hox gene transcript during embryonic development.

High difference can be noticed between E9.5 and E10.5, for which only 2 genes are misexpressed. This could be discussed more in depth in particular as E10.5 is chosen for the following ChIPseq study.

Thank you for your comment. As described above, although only 2 genes were statistically different ($\text{padj} < 0.05$), we confirmed the tendency of decreased expression of posterior Hox genes in *Kdm7a*^{-/-} embryos at E10.5 even during the RNA-seq analysis of E10.5 embryos (revised Fig. 2f). In addition, qPCR analysis demonstrated the downregulation of posterior Hox genes with statistical significance (revised Fig. 2g). In general, qPCR analysis is relatively more sensitive than RNA-seq analysis (in case of normal read depth) since the procedure of qPCR analysis includes amplification of targeted region. We believe that Hox genes were significantly down-regulated in *Kdm7a*^{-/-} embryos even at E10.5. In addition, chromatin immunoprecipitation

(ChIP) analysis requires large number of cells especially for repressive histone mark H3K9me2 and H3K27me2, as described in the original manuscript. That is why we have chosen E10.5 embryo despite the fact that Hox genes in E9.5 embryos were more sensitive to Kdm7a mutant than those in E10.5.

Why did the author not perform the comparison between head and trunk too to confirm the differential expression?

Thank you for your question. In this study, we aimed to examine whether Kdm7a impacts transcription during mouse development. The comparison of transcript between head and trunk is out of our study scope.

Detailed analysis of ChIP-seq:

Idem as RNAseq for replicates description & validation

Thank you for your comments and suggestions. Accordingly, we have performed ChIP-seq analysis with two biological replicates and confirmed that our analysis was highly correlated (please see revised supplementary Fig. 4). Then, we have recalculated the average log2 fold change of ChIP-seq signals based on READ COUNTS of H3K9me2 for each Hox gene, and visualized them as heatmaps (Please see revised Fig. 3d (KO/WT) and Supplementary Fig 5d (Head/trunk)). Briefly, we calculated log2 fold change of ChIP-seq signals using the following formula:

$$\log_2 \left\{ \left(\frac{\text{CPM}^{\text{IP1}}}{\text{CPM}^{\text{input1}}} \right) / \left(\frac{\text{CPM}^{\text{IP2}}}{\text{CPM}^{\text{input2}}} \right) \right\}$$

where CPM^{IP} and $\text{CPM}^{\text{input}}$ represent the read counts per million mapped reads (CPM) from IP and input libraries, respectively; we have provided detailed information in the Methods section, (Page 20, Line 619-640).

The comparison between head and trunk is a bit confusing. The biological question might be clarified to help the reader. It sounds like a characterization of the Hox inactive locus in the head versus active in the trunk.

Page 7, line 221-22: the sentence is confusing. It is not an increase of H3K9me2 but a higher enrichment of H3K9me2 in the head compared to the trunk, that is in line with the inactive state of the Hox genes in the head (and not in the trunk). Reformulating the sentence would be helpful for the clarity of the message driven by the data.

Thank you for your comment and we apologize for the lack of clarity. Accordingly, we have

revised the sentence to help readers to better understand the biological questions we posed and our message driven by the data, (Page 7, Line 225-251).

In line Fig 3D: somite/brain should be also related to head/trunk for helping the reader.

Thank you for your comment. We have rephrased “somite” and “brain” to “trunk” and “head”, respectively in revised supplementary Fig. 5e.

Moreover, interpreted conclusions could be added in the text for guiding the reader

Thank you for your comment. Alternative to adding the interpreted conclusions as text, we have provided a graphical model to summarize our findings to help readers to better understand our research summary. Please see revised Fig. 4.

-The state of H3K9me2 is enriched in the head if the genes are already downregulated/not expressed. This might be relevant to link this point with the choice of not doing the RNAseq on the head.

Thank you for your comment. As you mentioned, “H3K9me2 is enriched in the head where Hox genes are already downregulated or not expressed” could be another reason as to why the we didn’t perform RNA-seq on the head, in addition to the reason already described above (i.e. out of study scope).

Fig 3F: a large number of anterior Hox genes present an increase of H3K9me2 enrichment. This might be discussed.

Thank you for your comment and suggestion. Although the presentation of the heatmap was improved by the replicate experiment and their average calculation (please see the revised Fig 3d), we observed an increased level of some anterior Hox genes. This might be due to technical difficulties of ChIP-seq analysis of repressive histone marks especially for H3K9me2 (i.e. relatively hard to detect the peaks). Otherwise, histone marks, other than H3K9me2, might play an important role in the transcriptional regulation of the anterior Hox genes. In ChIP-qPCR analysis, we confirmed higher enrichment of H3K9me2 in *Kdm7a*^{-/-} embryos at *Hoxa13* locus in comparison with the *Hoxa3* locus. Thus, we believe that the posterior Hox genes could be preferentially regulated by H3K9me2 compared to the anterior Hox genes. Nevertheless, further detailed studies are warranted to elucidate the mechanisms through which anterior Hox genes

are regulated.

Figure 3D and 3H:

The error bars for the ChIP-qPCR are required. It seems that H3K4me3 is more enriched for the mutant compared to control for *Hoxa13* (which is in opposition with H3K9me2 repressive mark accumulation). It might be useful to change the normalization to Histone3 in this way.

Thank you for your comment. We have included additional ChIP-qPCR experiments and calculated the error bars. As you could see the revised Fig. 3e, H3K9me2 level at *Hoxa13* loci was significantly increased in the *Kdm7a*^{-/-} mice compared to wild-type ($P < 0.038$). Also, the difference in the H3K4me3 levels at *Hoxa13* loci was not statistically significant between the wild-type and the *Kdm7a* mutant. In addition, we have performed ChIP-qPCR analysis of total H3 and confirmed no statistical difference between the wild-type and the *Kdm7a* mutant at the *Actb*, *Hoxa3*, as well as *Hoxa13* loci (Please see revised Fig. 3e).

H3K27me2 regulation:

Finally, concerning the role of *Kdm7a* on K27me2, some additional experiments, at least by ChIP-qPCR on selected Hox targets (*Hox3*, *Hox13*...) and controls (at least two, known positive & negative) should be provided (KOsControl).

Thank you for your comment. Accordingly, we have performed ChIP-qPCR and ChIP-seq for H3K27me2. As the revised Fig. 3 and supplementary Fig. 5 show, enrichment of H3K27me2 was not altered between the wild-type and *Kdm7a*^{-/-} developmental trunk or between the head (where Hox genes are inactive) and the trunk (where Hox genes are active). Therefore, we believe that H3K9me2, but not H3K27me2, could be associated with the transcriptional state of Hox genes during mouse development. We have included these results in the revised manuscript.

Minor comments

Page 4, line138 & figure 1: Quantification of signal generated upon expression of sgRNA868/867 in HeLa cells control experiments could be provided

Thank you for your comment. We have added the experiment's quantification data for the sgRNA selection. Please see revised supplementary Fig. 1a.

Figure1D, RFLP: legend of mice genetic background is missing

Thank you for your comment and we apologize for the missing information. We have added the information regarding the genetic background. Please see the Figure Legend of the revised supplementary Fig. 1b.

Figure 2A: expression pattern of the Hox might be drawn on the cartoon to correlate with the embryos cutting and help the reader in the lecture

Thank you for your suggestion. We have sketched the expression pattern of Hox genes on the depiction in the revised Fig. 2a.

Figure 2D left panel second graphical view: Orthography: “comoplex”

Thank you for your comment. Accordingly, we have rephrased “comoplex” to “complex”.

Figure 3F: the case of Hox12c might be discussed

Thank you for your comment. As described above, we have performed the replicate experiment and improved the presentation of the heatmap including the case of Hoxc12. However, we could still observe the discrepancies between the Hox gene expression and their H3K9me2 marks. This might be due to technical difficulties in repressive histone mark or H3K9me2 independent (or additionally) regulatory mechanisms as we mentioned above. Further studies are warranted.

Figure 3F: It would help the reader if anterior and posterior Hox are indicated.

Thank you for your suggestion, we have added an indicator of anterior and posterior to the heatmap. In addition, we have asterisked Hox genes that were significantly downregulated as shown by the qPCR analysis in the E10.5 *Kdm7a* mutant embryos. Please see the revised Fig. 3d.

Figure 3: H3K9me2 and K4me3 should be normalized to H3 or at least, H3 ChIP-qPCR could be shown

Thank you for your comment. As described above, we have performed ChIP-qPCR analysis of the total H3 and confirmed no statistical difference between the wild-type and the *Kdm7a* mutant at *Actb*, *Hoxa3*, as well as *Hoxa13* loci (Please see the revised Fig. 3e). We have

included these results in the revised manuscript.

Page9, line297-301: it seems not in line with the result showing that Kdm7a KO is associated with a decrease of Hox gene expression. The result suggests that Kdm7a is an activator. The hypothesis developed links the repressive mark K27me3 and nuclear active environment that seems confusing. Reference should be added as well as the idea behind clarified.

Thank you for your comment. Since the concept regarding “Page9, Line297-301, paragraph 4 in discussion” was not reflected of our results as the reviewer pointed out, we have removed it from the text. Alternatively, we have added a discussion regarding H3K27me2, (Page 10, Line 322-338).

Page9, line 302-315: this part is based on assumption and should be nuanced. The link between the different studies needs to be re-enforced. Alternatively, this part may be removed as it sounds lacking strength.

Thank you for your suggestion. We have removed “Page9, Line302-315, paragraph 5 in discussion”.

- 1 Tsukada, Y., Ishitani, T. & Nakayama, K. I. KDM7 is a dual demethylase for histone H3 Lys 9 and Lys 27 and functions in brain development. *Genes & development* **24**, 432-437, doi:10.1101/gad.1864410 (2010).
- 2 Soshnikova, N. & Duboule, D. Epigenetic temporal control of mouse Hox genes in vivo. *Science* **324**, 1320-1323, doi:10.1126/science.1171468 (2009).
- 3 Izpisua-Belmonte, J. C., Falkenstein, H., Dolle, P., Renucci, A. & Duboule, D. Murine genes related to the Drosophila AbdB homeotic genes are sequentially expressed during development of the posterior part of the body. *EMBO J* **10**, 2279-2289 (1991).
- 4 Huang, C. *et al.* Dual-specificity histone demethylase KIAA1718 (KDM7A) regulates neural differentiation through FGF4. *Cell research* **20**, 154-165, doi:10.1038/cr.2010.5 (2010).
- 5 Deschamps, J. & van Nes, J. Developmental regulation of the Hox genes during axial morphogenesis in the mouse. *Development* **132**, 2931-2942, doi:10.1242/dev.01897 (2005).
- 6 Doetschman, T. Influence of genetic background on genetically engineered mouse phenotypes. *Methods in molecular biology* **530**, 423-433,

doi:10.1007/978-1-59745-471-1_23 (2009).

- 7 Kuroki, S. *et al.* Epigenetic regulation of mouse sex determination by the histone demethylase Jmjd1a. *Science* **341**, 1106-1109, doi:10.1126/science.1239864 (2013).
- 8 Xu, Y. *et al.* WERAM: a database of writers, erasers and readers of histone acetylation and methylation in eukaryotes. *Nucleic acids research* **45**, D264-D270, doi:10.1093/nar/gkw1011 (2017).

REVIEWERS' COMMENTS:

Reviewer #1 (Remarks to the Author):

The authors have addressed all my concerns. I suggest to accept.

I have two more minor points below. If the authors could make changes then no need to send back to me again.

In line 216 "...that Jmjd3 regulates Hox gene expression transcript but not boundaries.." The "boundaries" here could be misleading, could authors find another word instead?

In line 250-251, "no differences in H3K27me2 between the head and trunk" should not lead to the conclusion "suggesting that H3K9me2, but not H3K27me2, could be associated with the transcriptional state of Hox gene locus during mouse embryonic development." The right conclusion should be "suggesting KDM7A does not regulate Hox genes through H3K27me2".

Reviewer #4 (Remarks to the Author):

The authors raised all of our concerns with precise point by point answer and major improvements. Notably, the figure 4 is highly appreciated in the revised manuscript.

Author's responses to the reviewer's comments

REVIEWERS' COMMENTS:

Reviewer #1 (Remarks to the Author):

The authors have addressed all my concerns. I suggest to accept.

I have two more minor points below. If the authors could make changes then no need to send back to me again.

We highly appreciate you reviewing our current manuscript again. We are very grateful for the reviewer's positive comments. In response to the additional comments, we have revised the paper. Our point-by-point responses to the reviewer's comments are as follows:

In line 216 “..that Jmjd3 regulates Hox gene expression transcript but not boundaries..” The “boundaries” here could be misleading, could authors find another word instead?

According to the reviewer's comments, we have replaced the sentence as “... that Jmjd3 regulates Hox gene expression levels but not its spatial distribution.”.

In line 250-251, "no differences in H3K27me2 between the head and trunk" should not lead to the conclusion “suggesting that H3K9me2, but not H3K27me2, could be associated with the transcriptional state of Hox gene locus during mouse embryonic development.” The right conclusion should be "suggesting KDM7A does not regulate Hox genes through H3K27me2".

We highly appreciate the reviewer's comments on this point. Accordingly, we have replaced the sentence as “suggesting that Kdm7a might not regulate Hox genes expression through H3K27me2-mediated mechanisms.”

Reviewer #4 (Remarks to the Author):

The authors raised all of our concerns with precise point by point answer and major improvements. Notably, the figure 4 is highly appreciated in the revised manuscript.

We highly appreciate you reviewing our current manuscript again and we are very grateful for the reviewer's positive comments.